# Scaling Up Exact Neural Network Compression by ReLU Stability

**Thiago Serra**
Bucknell University
Lewisburg, PA, United States
`thiago.serra@bucknell.edu`

**Xin Yu**
University of Utah
Salt Lake City, UT, United States
`xin.yu@utah.edu`

**Abhinav Kumar**
Michigan State University
East Lansing, MI, United States
`kumarab6@msu.edu`

**Srikumar Ramalingam**
Google Research
New York, NY, United States
`rsrikumar@google.com`

## Abstract

We can compress a rectifier network while exactly preserving its underlying functionality with respect to a given input domain if some of its neurons are stable. However, current approaches to determine the stability of neurons with Rectified Linear Unit (ReLU) activations require solving or finding a good approximation to multiple discrete optimization problems. In this work, we introduce an algorithm based on solving a single optimization problem to identify all stable neurons. Our approach is on median 183 times faster than the state-of-art method on CIFAR-10, which allows us to explore exact compression on deeper ($5 \times 100$) and wider ($2 \times 800$) networks within minutes. For classifiers trained under an amount of $\ell_1$ regularization that does not worsen accuracy, we can remove up to 56% of the connections on CIFAR-10 dataset. The code is available at the following link, `https://github.com/yuxwind/ExactCompression`.

## 1  Introduction

For the past decade, the computing requirements associated with state-of-art machine learning models have grown faster than typical hardware improvements [5]. Although those requirements are often associated with training neural networks, they also translate into larger models, which are challenging to deploy in modest computational environments, such as in mobile devices.

Meanwhile, we have learned that the expressiveness of the models associated with neural networks—when measured in terms of their number of linear regions —grows polynomially on the number of neurons and occasionally exponentially on the network depth [69, 65, 73, 81, 36, 37]. Hence, we may wonder if the pressing need for larger models could not be countered by such gains in model complexity. Namely, if we could not represent the same model using a smaller neural network. More specifically, we consider the following definition of equivalence [49, 79]:

**Definition 1.** *Two neural networks $\mathcal{N}_1$ and $\mathcal{N}_2$ with associated functions $\boldsymbol{f}_1 : \mathbb{R}^{n_0} \to \mathbb{R}^m$ and $\boldsymbol{f}_2 : \mathbb{R}^{n_0} \to \mathbb{R}^m$ are local equivalent with respect to a domain $\mathbb{D} \subseteq \mathbb{R}^{n_0}$ if $\boldsymbol{f}_1(x) = \boldsymbol{f}_2(x) \, \forall x \in \mathbb{D}$.*

There is an extensive literature on methods for compressing neural networks [18, 11], which is aimed at obtaining smaller networks that are nearly as good as the original ones. These methods generally produce networks that are not equivalent, and thus require retraining the neural network for better accuracy. They may also lead to models in which the relative accuracy for some classes is more affected than that of other classes [43].

35th Conference on Neural Information Processing Systems (NeurIPS 2021).

Compressing a neural network while preserving its associated function is a relatively less explored topic, which has been commonly referred to as *lossless compression* [79, 83]. However, that term has also been used for the more general case in which the overall accuracy of the compressed network is no worse than that of the original network regardless of equivalence [95]. Hence, we regard *exact compression* as a more appropriate term when equivalence is preserved.

Exact compression has distinct benefits and challenges. On the one hand, there is no need for retraining and no risk of disproportionately affecting some classes more than others. On the other hand, optimization problems that are formulated for exact compression need to account for any valid input as opposed to relying on a sample of inputs. In this paper, we focus on how to scale such an approach to a point in which exact compression starts to become practical for certain applications.

In particular, we introduce and evaluate a faster algorithm for exact compression based on identifying all neurons with Rectified Linear Unit (ReLU) activation that have linear behavior, which are denoted as *stable*. In other words, those are the neurons for which the mapping of inputs to outputs is always characterized by a linear function, which is either the constant value $0$ or the preactivation output. We can remove or merge such neurons—and even entire layers in some cases—while obtaining a smaller but equivalent neural network. Our main contributions are the following:

(i) We propose the algorithm `ISA` (Identifying Stable Activations), which is based on solving a single Mixed-Integer Linear Programming (MILP) formulation to verify the stability of all neurons of a feedforward neural network with ReLU activations. `ISA` is faster than solving MILP formulations for every neuron—either optimally [90] or approximately [79]. Compared to [79], the median improvement is of **83** times on MNIST dataset (**183** times on CIFAR-10 dataset and **137** times on CIFAR-100 dataset) —and in fact greater in larger networks.

(ii) We reduce the runtime with a GPU-based preprocessing step that identifies neurons that are not stable with respect to the training set. The median improvement for that part alone is of **3**.**2** times on MNIST dataset.

(iii) We outline and prove the correctness of a new compression algorithm, `LEO++` (Lossless Expressiveness Optimization, as in [79]), which leverages (i) to perform all compressions once per layer instead of once per stable neuron [79].

(iv) We leverage the scalability of our approach to investigate exact compressibility on classifiers that are deeper ($5 \times 100$) and wider ($2 \times 800$) than previously studied in [79] ($2 \times 100$). We show that approximately $20\%$ of the neurons and $40\%$ of the connections can be removed from MNIST classifiers trained with an amount of $\ell_1$ regularization that does not worsen accuracy.

## 2 Related work

There are many pruning methods for sparsifying or reducing the size of neural networks by removing connections, neurons, or even layers. They are justified by the significant redundancy among parameters [23] and the better generalization bounds of compressed networks [8, 98, 87, 88].

Surveys such as [11] note that these methods are typically based on a tradeoff between model efficiency and quality: the models of compressed neural networks tend to have a comparatively lower accuracy, save some exceptions [35, 95, 87]. Nevertheless, the loss in accuracy due to compression is disproportionately distributed across classes and more severe in a fraction of them; the most impacted inputs are those that the original network could not classify well; and the overall robustness to noise or adversarial examples is diminished [43]. Furthermore, the amount of compression yielding similar performance to the original network can vary significantly depending on the task [59].

To make up for model changes and potential accuracy loss, one may rely on a three-step procedure consisting of (1) training the neural network; (2) compression; and (3) retraining. Nevertheless, the scope of compression methods is seldom restricted to the second step. For example, the compressibility of a neural network hinges on how it was trained, with regularizations such as $\ell_1$ and $\ell_2$ often used to make part of the network parameters negligible in magnitude—and hopefully in impact as well.

In fact, the two main—and recurring—themes in this topic are pruning connections when the corresponding parameters are sufficiently small [38, 66, 47, 35, 34, 57, 29, 31, 89] and when the impact of the connection on the loss function is sufficiently small [54, 39, 52, 64, 25, 96, 97, 56, 91, 92, 82]. The main issue with the first approach is that small weights may nevertheless be important,

although it is possible to empirically quantify their impact on the loss function [76]. The main issue with the second approach is the computational cost of calculating the second-order derivatives of the loss function in deep networks, which has lead to many approaches for approximating such values.

Overlapping with such approximations, there is a growing literature on casting neural network compression as an optimization problem [41, 62, 1, 96, 79, 26]. Most often, these formulations aim to minimize the impact of the compression on how the neural network performs on the training set.

Other lines of work and overlapping themes in neural network compression include combining similar neurons [84, 63, 86, 87]; low-rank approximation, factorization, and random projection of the weight matrices [46, 24, 51, 8, 93, 85, 91, 87, 88, 58]; and statistical tests on the relevance of a connection to network output [95]. Many recent approaches focus on pruning at initialization instead of after training [56, 55, 92, 89, 30] as well as on what parameters to use when these networks are retrained [29, 61, 74].

Exact compression was only recently explored for fully-connected feedforward neural networks [79] and graph neural networks [83]. Nevertheless, we may associate it with the literature on neural network equivalency, which includes verifying that networks are equivalent [67, 15], identifying operations that produce equivalent networks [42, 16, 50, 49, 72], reconstructing networks from their outputs [3, 4, 27, 2, 75], and evaluating the effect of redundant representations on training [10, 71].

## 3 Setting and notation

We consider fully-connected feedforward neural networks with $L$ hidden layers, in which we denote $n_l$ as the number of units—or width—of layer $l \in \mathbb{L} := \{1, 2, \ldots, L\}$ and $x_i^l$ as the output of the $i$-th unit of layer $l$ for $i \in \{1, 2, \ldots, n_l\}$. For uniformity, we denote $\boldsymbol{x}^0 \in \mathbb{R}^{n_0}$ as the network input. We denote the output of the $i$-th unit of layer $l$ as $x_i^l = \sigma^l(y_i^l)$, where the pre-activation output $y_i^l := \boldsymbol{w}_i^l \cdot \boldsymbol{x}^{l-1} + b_i^l$ is defined by the learned weights $\boldsymbol{w}_i^l \in \mathbb{R}^{n_{l-1}}$ and the bias $b_i^l \in \mathbb{R}$ of the unit as well as the activation function $\sigma^l : \mathbb{R} \to \mathbb{R}$ associated with layer $l$, which is $\sigma^l(u) = \max\{0, u\}$—the ReLU [33]. The output layer may have a different structure, such as the softmax layer [13], which is nevertheless irrelevant for our purpose of compressing the hidden layers. For every layer $l \in \mathbb{L}$, let $\boldsymbol{W}^l = [\boldsymbol{w}_1^l \boldsymbol{w}_2^l \ldots \boldsymbol{w}_{n_l}^l]^T$ be the matrix of weights, $\boldsymbol{W}_{\mathbb{S}}^l$ be a submatrix of $\boldsymbol{W}^l$ consisting of the rows in set $\mathbb{S}$, and $\boldsymbol{b}^l = [b_1^l b_2^l \ldots b_{n_l}^l]^T$ be the vector of biases. Finally, let $\boldsymbol{I}_m(\mathbb{S})$ be an $m \times m$ diagonal matrix in which the $i$-th diagonal element is 1 if $i \in \mathbb{S}$ and 0 if $i \notin \mathbb{S}$.

## 4 Identifying stability for exact compression

This section explains the concept of stability and describes how MILP has been used to identify stable neurons. If the output of neuron $i$ in layer $l$ is always linear on its inputs, we say that the neuron is stable. This happens in two ways for the ReLU activation. When $x_i^l = 0$ for any valid input, which implies that $y_i^l \leq 0$, we say that the neuron is *stably inactive*. When $x_i^l = y_i^l$ for any valid input, which implies that $y_i^l \geq 0$, we say that the neuron is *stably active*.

The qualifier *valid* is essential since not every input may occur in practice. If $\boldsymbol{w}_i^l \neq \boldsymbol{0}$, there are nonempty halfspaces on $\boldsymbol{x}^{l-1}$ that would make that neuron active or inactive, $\{\boldsymbol{x}^{l-1} : \boldsymbol{w}_i^l \cdot \boldsymbol{x}^{l-1} + b_i^l \leq 0\}$ and $\{\boldsymbol{x}^{l-1} : \boldsymbol{w}_i^l \cdot \boldsymbol{x}^{l-1} + b_i^l \geq 0\}$, but it is possible that valid inputs only map to one of them. For the first layer, we only need to account for the valid inputs to the neural network. For example, the domain of a network trained on the MNIST dataset is $\{\boldsymbol{x}^0 : \boldsymbol{x}^0 \in [0, 1]^{784}\}$ [53]. For the remaining hidden layers, we also account for the combinations of outputs that can be produced by the preceding layers given their valid inputs and parameters. Hence, assessing stability is no longer straightforward.

We can determine if a neuron of a trained neural network is stable by solving optimization problems to maximize and minimize its preactivation output [90]. The main decision variables in these problems are the inputs for which the preactivation output is optimized. Consequently, there is also a decision variable associated with the output of every neuron, in addition to other variables described below.

**MILP formulation of a single neuron**  For every neuron $i$ of layer $l$, we map every input vector $\boldsymbol{x}^{l-1}$ to the corresponding output $x_i^l$ through a set of linear constraints that also include a binary variable $z_i^l$ denoting if the unit is active or not, a variable for the pre-activation output $y_i^l$, a variable $\chi_i^l := \max\{0, -y_i^l\}$ denoting the output of a complementary fictitious unit, and positive constants

$M_i^l$ and $\mu_i^l$ that are as large as $x_i^l$ and $\chi_i^l$ can be. The formulation below is explained in Appendix A1.

$$\boldsymbol{w}_i^l \cdot \boldsymbol{x}^{l-1} + b_i^l = y_i^l = x_i^l - \chi_i^l \tag{1}$$

$$0 \le x_i^l \le M_i^l z_i^l \tag{2}$$

$$0 \le \chi_i^l \le \mu_i^l(1 - z_i^l) \tag{3}$$

$$z_i^l \in \{0, 1\} \tag{4}$$

**Using MILP to determine stability**    Let $\mathbb{X} \subset \mathbb{R}^{n_0}$ be the set of valid inputs for the neural network, which we may assume to be bounded in every direction. We can obtain the interval $[\underline{\mathcal{Y}}_i^{l'}, \overline{\mathcal{Y}}_i^{l'}]$ for the preactivation output $y_i^{l'}$ of neuron $i$ in layer $l'$ by solving the following optimization problems [90]:

$$\underline{\mathcal{Y}}_i^{l'} := \left\{ \min \boldsymbol{w}_i^{l'} \cdot \boldsymbol{x}^{l'-1} + b_i^{l'} : \boldsymbol{x}^0 \in \mathbb{X}; (1) - (4) \ \forall l \in [l'-1], i \in [n_l] \right\} \tag{5}$$

$$\overline{\mathcal{Y}}_i^{l'} := \left\{ \max \boldsymbol{w}_i^{l'} \cdot \boldsymbol{x}^{l'-1} + b_i^{l'} : \boldsymbol{x}^0 \in \mathbb{X}; (1) - (4) \ \forall l \in [l'-1], i \in [n_l] \right\} \tag{6}$$

When $\overline{\mathcal{Y}}_i^{l'} \le 0$, then $x_i^{l'} = 0$ for every $x^0 \in \mathbb{X}$ and the neuron is stably inactive. When $\underline{\mathcal{Y}}_i^{l'} \ge 0$, then $x_i^{l'} = y_i^{l'}$ for every $x^0 \in \mathbb{X}$ and the neuron is stably active.

Variations of the formulations above have been proposed for diverse tasks over neural networks, such as verifying them [17], embedding their model into a broader decision-making problem [78, 9, 21], and measuring their expressiveness [81]. When stable units are identified, other optimization problems over trained neural networks become easier to solve [90]. For example, weight regularization can induce neuron stability and facilitate adversarial robustness verification [94]. There is extensive work on the properties of such formulations and methods to solve them more effectively [28, 7, 12, 80, 6].

For the purpose of identifying stable neurons, however, it is not scalable to analyze large neural networks by solving such optimization problems for every neuron [90]—or even by just approximately solving each of them to ensure that $\overline{\mathcal{Y}}_i^{l'} \le 0$ or $\underline{\mathcal{Y}}_i^{l'} \ge 0$ [79].

## 5 A new algorithm for exact compression

Based on observations discussed in what follows (I to III), we propose a new MILP formulation to identify stable neurons (Section 5.1), means to generate feasible solutions while the formulation is solve (Section 5.2), a preprocessing step to reduce the effort to solve the formulation (Section 5.3), the resulting algorithm ISA for identifying all stable neurons at once (Section 5.4), and the revised compression algorithm LEO++ exploiting all stable neurons in each layer at once (Appendix A5).

### 5.1 A new MILP formulation

Consider the two observations below and their implications:

**I: The overlap between optimization problems**    Although previous approaches require solving many optimization problems, their formulations are all very similar: we maximize or minimize the same objective function for each neuron, the feasible set of the problems for each layer are the same, and they are contained in the feasible set of problems for the subsequent layers.

**II: Proving stability is harder than disproving it**    Certifying that a neuron is stable is considerably more complex than certifying that a neuron is *not* stable. For the former, we need to exhaustively show that all inputs lead to the neuron always being active or always being inactive, which can be achieved by solving (5) and (6) for every neuron. For the latter, we just need a pair of inputs to the neural network such that the neuron is active with one of them and inactive with the other.

Therefore, we consider the problem of finding an input that serves as a certificate of a neuron not being stable to as many neurons of unknown classification as possible. For that purpose, we define a decision variable $p_i^l \in \{0, 1\}$ to denote if an input activates neuron $i$ in layer $l$. Likewise, we define a decision variable $q_i^l \in \{0, 1\}$ to denote if an input does not activate neuron $i$ in layer $l$. Furthermore, we restrict the scope of the problem to state that have not been previously observed

by using $\mathbb{P}^l \subseteq \{1, \ldots, n_l\}$ as the set of neurons in layer $l$ for which there is no known input that activates the neuron. Likewise, we use $\mathbb{Q}^l \subseteq \{1, \ldots, n_l\}$ as the set of neurons in layer $l$ for which there is no known input that does not activate the neuron. For brevity, let $\boldsymbol{P} := (\mathbb{P}^1, \ldots, \mathbb{P}^L)$ and $\boldsymbol{Q} := (\mathbb{Q}^1, \ldots, \mathbb{Q}^L)$ characterize an instance of such optimization problem, which is formulated as follows:

$$\mathcal{C}(\boldsymbol{P}, \boldsymbol{Q}) = \max \quad \sum_{l \in \mathbb{L}} \left( \sum_{i \in \mathbb{P}^l} p_i^l + \sum_{i \in \mathbb{Q}^l} q_i^l \right) \tag{7}$$

$$\text{s.t.} \quad \boldsymbol{x}^0 \in \mathbb{X} \tag{8}$$

$$(1) - (4) \ \forall l \in \mathbb{L}, i \in [n_l] \tag{9}$$

$$0 \le p_i^l \le z_i^l \ \forall l \in \mathbb{L}, i \in \mathbb{P}^l \tag{10}$$

$$0 \le q_i^l \le 1 - z_i^l \ \forall l \in \mathbb{L}, i \in \mathbb{Q}^l \tag{11}$$

$$p_i^l, q_i^l \in \{0, 1\} \tag{12}$$

Note that constraint (12) is actually not necessary. We refer to Appendix A2 for more details.

The formulation above yields an input that maximizes the number of neurons with an activation state that has not been previously observed. The following results show that it entails an approach in which no more than $N + 1$ such formulations are solved. We refer to Appendix A3 for the proofs.

**Proposition 1.** *If $\mathcal{C}(\boldsymbol{P}, \boldsymbol{Q}) = 0$, then every neuron $i \in \mathbb{P}^l$ is stably inactive and every neuron $i \in \mathbb{Q}^l$ is stably active.*

**Corollary 2.** *The stability of all neurons of a neural network can be determined by solving formulation (7)–(12) at most $N + 1$ times, where $N := \sum_{l \in \mathbb{L}} n_l$.*

Those results imply that we can iteratively solve the new formulation as part of an algorithm to identify all stable neurons. In fact, we can determine the stability of the entire neural network with a single call to the MILP solver. Except for the last time that formulation (7)–(12) is solved, there is no need to solve it to optimality: any solution with a positive objective function value can be used to reduce the number of unobserved states. Hence, all that we need is a way to inspect every feasible solution obtained by the MILP solver and then remove the solutions in which either $p_i^l = 1$ or $q_i^l = 1$ for states that were already observed. Both of those needs can be addressed in fully-fledged MILP solvers by implementing a lazy constraint callback. We refer to Appendix A4 for more details. When we finally reach $\mathcal{C}(\boldsymbol{P}, \boldsymbol{Q}) = 0$, the correctness of the MILP solver serves as a certificate of the stability of those remaining neurons. The resulting algorithm is described in Section 5.4.

## 5.2 Inducing feasible MILP solutions

The runtime with a single solver call depends on the frequency with which feasible solutions are obtained. Although at most $N + 1$ optimal solutions would suffice if we were to make consecutive calls to the solver until $\mathcal{C}(\boldsymbol{P}, \boldsymbol{Q}) = 0$, we should not expect the same from the first $N + 1$ feasible solutions found by the MILP solver while using the lazy constraint callback because they may not have a positive objective function value due to the $p_i^l$ and $q_i^l$ variables that have been fixed to 0. On top of that, obtaining a feasible solution for an MILP formulation is NP-complete [19].

**III: Finding feasible solutions to MILP formulations of neural networks is easy** To any valid input of the neural network there is a corresponding solution of the MILP formulation: the neural network input implies which neurons are active and what is their output when active [28].

Although any random input would suffice, we have found that it is better in practice to use inputs indirectly generated by the MILP solver. Namely, we can use the solution of the Linear Programming (LP) relaxation, which is solved at least once per branch-and-bound node. The LP relaxation is obtained from the MILP formulation by relaxing its integrality constraints. In the case of binary variables with domain $\{0, 1\}$, that consists of relaxing the domain of such variables to the continuous interval $[0, 1]$. We use the values of $\boldsymbol{x}^0$ in the solution of the LP relaxation as the network input, and thus obtain a feasible MILP solution by replacing the values of the other variables—which may be fractional for the decision variables with binary domains—by the values implied by fixing $\boldsymbol{x}^0$. However imprecise due to the relaxation of the binary domains, the input defined by the optimal solution of the LP relaxation may intuitively guide us toward maximizing the objective function.

## 5.3 Preprocessing

In addition to generating feasible MILP solutions at every node of the branch-and-bound tree during the solving process, we also evaluate the training set on the trained neural network to reduce the number of states that need to be search for by the MILP solver. By using GPUs, this step can be completed in few seconds for all the experiments performed.

## 5.4 Identifying stable neurons

Algorithm 1, which we denote ISA (Identifying Stable Activations), identifies all stable neurons of a neural network. The prior discussion on identifying stable units leads to the steps described between lines 3 and 28. First, $P$ and $Q$ are initialized between lines 3 and 6 according to the preprocessing step described above. Next, the MILP formulation is iteratively solved between lines 7 and 28. The block between lines 8 and 9 identifies the termination criterion, which implies that the unobserved states cannot be obtained with any valid input. The block between lines 10 and 24 inspects every feasible solution to identify unobserved states and then to effectively remove the decision variables associated with those states from the objective function by adding a constraint that sets their value to 0. The block between lines 25 and 26 produces a feasible solution from a solution of the LP relaxation when the latter is produced by the MILP solver. For brevity, we assume that the block between lines 10 and 24 would leverage such solution at the next repetition of the loop.

---

**Algorithm 1** ISA provably identifies all stable neurons of a neural network by iterating over the solution of a single MILP formulation to verify the occurrence of states unobserved in the training set

---

1: **Input:** neural network $\left(L, \left\{(n_l, \boldsymbol{W}^l, \boldsymbol{b}^l)\right\}_{l\in\mathbb{L}}\right)$
2: **Output:** stable neurons $\left(\left\{(\mathbb{P}^l, \mathbb{Q}^l)\right\}_{l\in\mathbb{L}}\right)$
3: **for** $l \leftarrow 1$ **to** $L$ **do**                                                     ▷ Pre-processing step
4:     $\mathbb{P}^l \leftarrow$ subset of $\{1, \ldots, n_l\}$ that is *never* activated by the training set
5:     $\mathbb{Q}^l \leftarrow$ subset of $\{1, \ldots, n_l\}$ that is *always* activated by the training set
6: **end for**
7: **while** solving $\mathcal{C}(\boldsymbol{P}, \boldsymbol{Q})$ **do**                           ▷ Loop interacting with MILP solver
8:     **if** optimal value is proven to be 0 **then**                       ▷ Remaining neurons are all stable
9:         **break**                                                         ▷ Nothing else to be done
10:     **else if** found positive MILP solution $(\bar{x}, \bar{z}, \bar{p}, \bar{q})$ **then**        ▷ Identified unobserved states
11:         **for** $l \leftarrow 1$ **to** $L$ **do**                                 ▷ Loops over all hidden layers
12:             **for every** $i \in \mathbb{P}^l$ **do**           ▷ Loops over neurons that have not been seen *active* yet
13:                 **if** $\bar{p}_i^l > 0$ **then**                               ▷ Neuron is active for the first time
14:                     $\mathbb{P}^l \leftarrow \mathbb{P}^l \setminus \{i\}$           ▷ Neuron is not stably inactive
15:                     $p_i^l \leftarrow 0$                       ▷ Restricts MILP to avoid identifying neuron again
16:                 **end if**
17:             **end for**
18:             **for every** $i \in \mathbb{Q}^l$ **do**           ▷ Loops over neurons that have not been seen *inactive* yet
19:                 **if** $\bar{q}_i^l > 0$ **then**                               ▷ Neuron is inactive for the first time
20:                     $\mathbb{Q}^l \leftarrow \mathbb{Q}^l \setminus \{i\}$           ▷ Neuron is not stably active
21:                     $q_i^l \leftarrow 0$                       ▷ Restricts MILP to avoid identifying neuron again
22:                 **end if**
23:             **end for**
24:         **end for**
25:     **else if** found LP relaxation solution $(\tilde{x}, \tilde{z}, \tilde{p}, \tilde{q})$ **then**     ▷ Input $\tilde{x}$ *may* produce unseen states
26:         use $\tilde{x}^0$ to produce an MILP solution $(\bar{x}, \bar{z}, \bar{p}, \bar{q})$   ▷ Produce unseen activations for input
27:     **end if**
28: **end while**
29: **return** $\left(\left\{(\mathbb{P}^l, \mathbb{Q}^l)\right\}_{l\in\mathbb{L}}\right)$

---

# 6 Experimental results

We trained and evaluated the compressibility of classifiers for the datasets MNIST [53], CIFAR-10 [48], and CIFAR-100 [48] with and without $\ell_1$ weight regularization, which is known to induce stability [90]. We refer to Appendix A6 for details on environment and implementation. We use the notation $L \times n$ for the architecture of $L$ hidden layers with $n$ neurons each. We started at $L = 2$ and $n = 100$, and then doubled the width $n$ or incremented the depth $L$ until the majority of the runs for MNIST classifiers for any configuration timed out after 3 hours. With preliminary runs, we chose values for $\ell_1$ which spanned from those for which accuracy is improving as $\ell_1$ increases until those for which the accuracy starts decreasing. We trained and evaluated neural networks with 5 different random initialization seeds for each choice of $\ell_1$. The amount of regularization used did not stabilize the entire layer. We refer to Appendix A7 for additional figures and tables with complete results.

**Regularization and compression**   Fig. 1 illustrates the average accuracy and number nodes that can be removed from networks according to architecture and dataset based on the amount of regularization used. When used in moderate amounts, regularization improves accuracy and very often that also coincides with enabling exact compression. We observe regularization improving accuracy in 17 of the 21 plots in Fig. 1. In 14 cases, we reduce the size of these more accurate networks. Those include all architectures for MNIST (a to g), the architecture with width 400 for CIFAR-100 (r), and all the architectures with more hidden layers for CIFAR-10 and CIFAR-100 (j, l, n, q, s, u).

**Runtime improvement**   Fig. 2 compares the baseline [79] with our approach on smaller MNIST classifiers—$2 \times 100$, $2 \times 200$, and $2 \times 400$—using $\ell_1$ as described above. The median ratio between runtimes is 100. The overall speedup is greater in larger networks: the median runtime ratio is 77 for $2 \times 100$, 153 for $2 \times 200$, and 193 for $2 \times 400$. By comparing the runtimes when not timing out with and without the preprocessing step in $2 \times 100$, we observe a median speed up of 3.2.

**Effect of regularization on compressibility**   We observe more compression with more $\ell_1$ regularization. For sufficiently large networks having the same accuracy as those trained with $\ell_1 = 0$ on MNIST, we can remove around 20% of the neurons and 40% of the connections. In line with [79], we observe that the exact compressibility of neural networks trained with $\ell_1 = 0$ is negligible, but also that you can have the cake and eat it too: certain choices of regularization lead to better accuracy and a smaller network. However,the cake can get very expensive as runtimes increase considerably.

**Relationship between compressibility and accuracy**   Fig. 3 analyzes the relationship between classifier accuracy and the number of neurons left after compression for $2 \times 100$ classifiers. When excluding uncompressible networks with $\ell_1 = 0$ or sufficiently small, we obtain a linear regressions with coefficient of determination ($R^2$) of 69% on MNIST, 91% on CIFAR-10, and 61% on CIFAR-100. That suggests that accuracy is a good proxy for how much a neural network can be compressed.

**Motivation for exact compression**   We also compared exact compression with Magnitude-based Pruning (MP), one of the most commonly used inexact methods. First, we identified all the connections that would be pruned by the removal of stably inactive neurons with our approach, which would also be harmless if identified and removed by MP. Second, we ranked all the connections based on the absolute value of their coefficients in order to identify at what pruning ratio those connections would have been removed by MP. We consistently found out across architectures of different sizes and levels of regularization that some of the pruned connections by our method would be found by MP at the 99[th] percentile. In other words, even though such connections would have no impact if removed from the network, MP would only resort to removing them at very extreme levels of pruning. Furthermore, if the same pruning ratio is used with MP, on average 10% of the total number of connections—or 18% of the pruned connections—removed by our method would not be removed by MP.

## 6.1 Limitations and alternatives

Due to the use of MILP solvers, our approach is not applicable to very large networks. In what follows, we consider ways to extend our approach by lifting some or all the guarantees provided.

**Inexact approach to larger networks**   We evaluated the impact of using only the quick preprocessing step described in Section 5.3 to determine which neurons to remove. Note that the preprocessing step is in principle intended to identify neurons that are not stable in order to avoid spending further time on them, but we can conversely assume that all the other neurons are stable at the cost of removing more than we should. By using preprocessing alone, we identified on average 31.93%

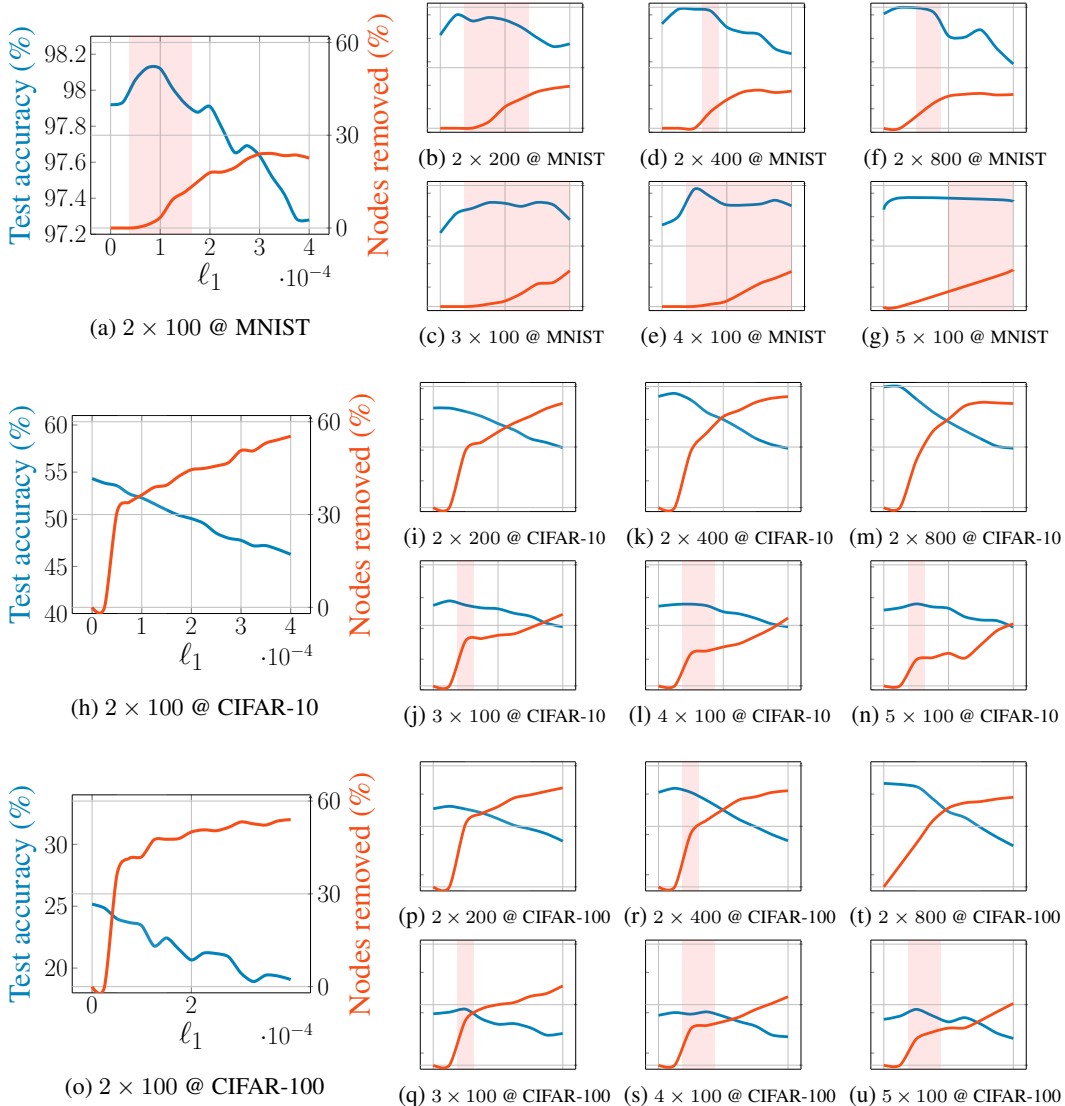

Figure 1: **Test accuracy** and **nodes removed** for varying amounts of $\ell_1$ **regularization.** The plots correspond to classifiers with different architectures on the (a)-(g) MNIST, (h)-(n) CIFAR-10, and (o)-(u) CIFAR-100 datasets. For each dataset, we keep the ranges of all the axes of the smaller plots same as the bigger plot but hide the ticks for brevity. Networks trained with $\ell_1$ regularization can be exactly compressed, even when regularization improves accuracy. In light red background, test accuracy is better than with no regularization (blue curve) and exact compression occurs (red curve).

potentially stable neurons in MNIST classifiers, $40.86\%$ in CIFAR-10 classifiers, and $41.98\%$ in CIFAR-100 classifiers. Among those neurons, only a few were actually not stable when evaluated with the test set. In terms of the number of not stable neurons with respect to the test set divided by the number of stable neurons with respect to the training set, we would have removed $1.16\%$ more neurons that we should for MNIST, $0.60\%$ for CIFAR-10, and $1.19\%$ for CIFAR-100. We refer to Appendix A7.5 for experiments involving convolutional neural networks (CNNs).

**Restricting exact compression to more likely inputs** We also tested the effect of bounding the sum of all the MNIST inputs to be within the minimum and maximum observed values. In particular, we have constrained the sum of all inputs to be within the interval $[15, 320]$ instead of $[0, 784]$. We believed that this approach would be preferable to constraining the value of individual inputs, since that would have affected the output upon rotation and translation. Note that this constraint is equivalent to imposing a prior on the number of foreground pixels on the digits to be within a range.

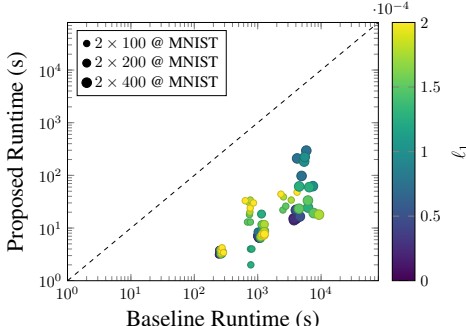
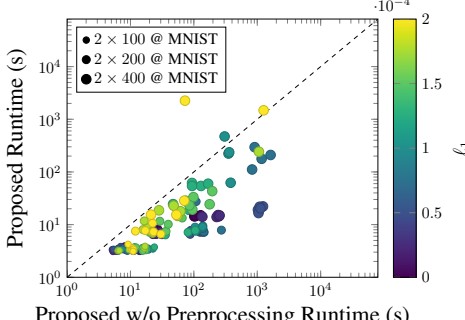

Figure 2: **Comparison of runtimes (in seconds) to identify all stable neurons.** On the left, we compare the proposed approach against the baseline from [79]. On the right, we compare the proposed approach with preprocessing against the proposed approach without preprocessing.

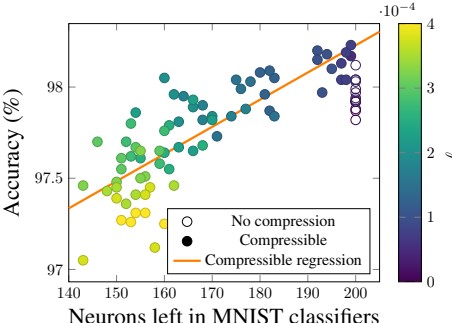
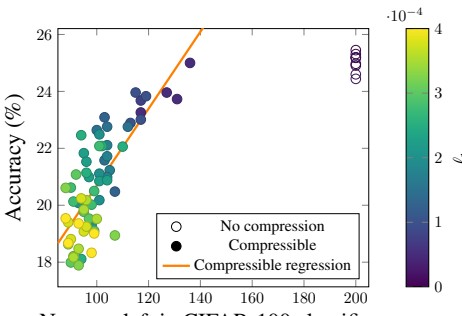

Figure 3: **Relationship between size of compressed neural networks and accuracy on** $2 \times 100$ **classifiers.** The coefficient of determination ($R^2$) for the linear regression obtained for accuracy based on neurons left for compressible networks is 69% on MNIST and 61% on CIFAR-100.

Global priors that jointly act on all the pixels have been used in computer vision in the pre-deep learning era, e.g., [22]. By restricting the analysis to the cases in which the time limit has not been exceeded either before or after the change, we obtained a better runtime in 69.6% of the cases and the runtime geometric mean went down by 17.7%.

# 7 Conclusion

This paper outlined the potential for exact compression of neural networks and presented an approach that makes it practical for sizes that are large enough for many applications. To the best of our knowledge, our approach is the state-of-the-art for optimization-based exact compression. Our performance improvements come from insights about the MILP formulations associated with optimization problems over neural networks, which have many other applications besides exact compression. Most notably, such formulations are also used for network verification [14, 60, 77].

**Societal Impact** Large models are resource-intensive for both training as well as inference. In contrast to approximate methods, our exact model compression algorithms can help deep learning practitioners to save computational time and resources without worrying about any loss in performance. That helps preventing the documented side effect of disproportionally degrading performance for some classes more than for other classes when the indicator of a successful compression is the overall performance, which could also lead to fairness issues [43, 68, 44].

**Acknowledgements** Thiago Serra was supported by the National Science Foundation (NSF) grant IIS 2104583. Xin Yu and Srikumar Ramalingam were partially funded by the NSF grant IIS 1764071. Abhinav Kumar was partially funded by the Ford Motor Company and the Army Research Office (ARO) grant W911NF-18-1-0330. We also thank the anonymous reviewers for their constructive feedback that helped in shaping the final manuscript.

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
