# Scaling Up Exact Neural Network Compression by ReLU Stability

# Supplementary Material

## A1    Description of MILP formulation for a ReLU activation

The formulation below is used to identify inputs for which a given output or activation pattern can be achieved. The decision variables include (i) the vector $\boldsymbol{x}^0$ associated with the input of the neural network; (ii) the vector $\boldsymbol{y}^l$ associated with the preactivation output of each hidden layer of the neural network; (iii) the vector $\boldsymbol{x}^l$ associated with the output of each hidden layer of the neural network; (iv) the vector $\chi^l$ associated with the complementary output of each hidden layer of the neural network; and (v) the binary vector $\boldsymbol{z}^l$ defining which neurons are active or not in each hidden layer of the neural network. The vector of weights $\boldsymbol{w}_i^l$ and the bias $b_i^l$ associated with each neuron as well as the constants $M_i^l$ and $\mu_i^l$ are coefficients of the formulation. The constraints are as follows:

$$\boldsymbol{w}_i^l \cdot \boldsymbol{x}^{l-1} + b_i^l = y_i^l \tag{13}$$

$$y_i^l = x_i^l - \chi_i^l \tag{14}$$

$$x_i^l \le M_i^l z_i^l \tag{15}$$

$$\chi_i^l \le \mu_i^l (1 - z_i^l) \tag{16}$$

$$x_i^l \ge 0 \tag{17}$$

$$\chi_i^l \ge 0 \tag{18}$$

$$z_i^l \in \{0, 1\} \tag{19}$$

Constraint (13) matches the layer input $\boldsymbol{x}^{l-1}$ with the neuron preactivation output $y_i^l$. We then use the binary variable $z_i^l$ to match $y_i^l$ with the neuron output with either $x_i^l$ or 0. When $z_i^l = 1$, constraints (16) and (18) imply that $\chi_i^l = 0$, and thus $x_i^l = y_i^l$ due to constraint (14). That only happens if $y_i^l \ge 0$ due to constraint (17). When $z_i^l = 0$, constraints (15) and (17) imply that $x_i^l = 0$, and thus $\chi_i^l = -y_i^l$. That only happens if $y_i^l \le 0$ due to constraint (18).

## A2    On dropping constraint (12)

We avoid explicitly enforcing that variables $p_i^l$ and $q_i^l$ are binary by leveraging that $z_i^l$ is binary. Constraint (10) implies that $p_i^l \in [0, 1]$ and $p_i^l \ne 0$ only if $z_i^l = 1$. In turn, if $z_i^l = 1$, then we can assume $p_i^l = 1$ by optimality since the objective function (7) maximizes the sum of those variables and no other constraint limits its value. Likewise, constraint (11) implies that $q_i^l \in [0, 1]$ and $q_i^l \ne 0$ only if $z_i^l = 0$. In turn, if $z_i^l = 0$, then likewise we can assume $q_i^l = 1$ by optimality since the objective function (7) maximizes the sum of those variables and no other constraint limits its value. Reducing the number of binary variables is widely regarded as a good practice to make MILP formulations easier to solve.

## A3    Proofs from Section 5.1

**Proposition 1.** *If $\mathcal{C}(\boldsymbol{P}, \boldsymbol{Q}) = 0$, then every neuron $i \in \mathbb{P}^l$ is stably inactive and every neuron $i \in \mathbb{Q}^l$ is stably active.*

*Proof.* Constraint (10) is the only upper bound on $p_i^l$ besides constraint (12). Hence, if there is any solution $(\bar{x}, \bar{z}, \bar{p}, \bar{q})$ of (9)–(12) in which $\bar{z}_i^l = 1$ for some $i \in \mathbb{P}_i^l, l \in \mathbb{L}$, then either $\bar{p}_i^l = 1$ or there is another solution $(\bar{\bar{x}}, \bar{\bar{z}}, \bar{\bar{p}}, \bar{\bar{q}})$ in which $\bar{\bar{p}}_i^l = 1$ and all other variables have the same value.

Likewise, constraint (10) is the only upper bound on $q_i^l$ besides constraint (12). Hence, if there is any solution $(\bar{x}, \bar{z}, \bar{p}, \bar{q})$ of (9)–(12) in which $\bar{z}_i^l = 0$ for some $i \in \mathbb{P}_i^l, l \in \mathbb{L}$, then either $\bar{q}_i^l = 1$ or there is another solution $(\bar{\bar{x}}, \bar{\bar{z}}, \bar{\bar{p}}, \bar{\bar{q}})$ in which $\bar{\bar{q}}_i^l = 1$ and all other variables have the same value.

If $\mathcal{C}(\boldsymbol{P}, \boldsymbol{Q}) = 0$, then for every solution $(\bar{x}, \bar{z}, \bar{p}, \bar{q})$ it follows that $\bar{p}_i^l = 0 \ \forall i \in \mathbb{P}^l, l \in \mathbb{L}$ and $\bar{q}_i^l = 0 \ \forall i \in \mathbb{Q}^l, l \in \mathbb{L}$, and consequently $\bar{z}_i^l = 0 \ \forall i \in \mathbb{P}^l, l \in \mathbb{L}$ and $\bar{z}_i^l = 1 \ \forall i \in \mathbb{Q}^l, l \in \mathbb{L}$. Thus, the neurons in $\mathbb{P}^l$ are always inactive and the neurons in $\mathbb{Q}^l$ are always active for any valid input. $\quad\square$

**Corollary 2.** *The stability of all neurons of a neural network can be determined by solving formulation* (7)–(12) *at most* $N + 1$ *times, where* $N := \sum\limits_{l \in \mathbb{L}} n_l$.

*Proof.* Let us initially consider a formulation in which $\mathbb{P}^l = \mathbb{Q}^l = \{1, \ldots, n_l\} \ \forall l \in \mathbb{L}$ and then respectively remove from those sets each neuron $i$ for which $p_i^l = 1$ and $q_i^l = 1$ in any solution obtained. When the formulation is first solved, we remove each neuron from either $\mathbb{P}^l$ or $\mathbb{Q}^l$, and therefore $N$ states remain unobserved. In subsequent steps, either (i) $\mathcal{C}(\boldsymbol{P}, \boldsymbol{Q}) > 0$ and the number of unobserved states decreases; or (ii) $\mathcal{C}(\boldsymbol{P}, \boldsymbol{Q}) = 0$, and thus any neuron $i \in \mathbb{P}^l$ is stably inactive and any neuron $i \in \mathbb{Q}^l$ is stably active. $\quad\square$

## A4   On lazy constraint callbacks

Lazy constraint callbacks are generally used when the total number of constraints of an MILP formulation is prohibitively large. One such example is the most commonly used formulation for the traveling salesperson problem due to the subtour elimination constraints [20]. The callback allows us to handle such cases more efficiently by formulating the problem with fewer constraints and then adding the remaining ones only if they are necessary to rule out infeasible solutions. Every time that a supposedly feasible solution is found, the MILP solver invokes the callback implemented by the user for an opportunity to make such a solution infeasible by adding one of the missing constraints that the supposedly feasible solution does not satisfy. If none is provided by the callback, the MILP solver accepts the solution as feasible.

In our case, we use a lazy constraint callback for a slightly different purpose. Namely, we implement the callback to (i) inspect every feasible solution that is obtained; and (ii) mimic the updates that would have been made to $\boldsymbol{P}$ and $\boldsymbol{Q}$ between consecutive calls to the solver by adding constraints that set the value of either $p_i^l$ or $q_i^l$ to zero once a solution is found in which such variable has a positive value. In other words, the callback adds constraints to ignore the effect of $p_i^l$ or $q_i^l$ on the objective function if we know that the $i$-th neuron of layer $l$ is active or inactive for some input, respectively. Therefore, the MILP solver will eventually produce an optimal solution of value zero once the set of solutions inspected by the callback covers all the possible states for the neurons and the remaining states are deemed unattainable after an exhaustive search.

## A5   A revised algorithm for compressing the neural network

Algorithm 2, which we denote as LEO++ (Lossless Expressiveness Optimization, as in [79]), leverages neuron stability for exactly compressing neural networks. We describe next each form of compression contained in the algorithm. For ease of explanation, they are in reverse order of appearance. These compression operations are the same as in [79], but performed once per layer instead of once per neuron. In comparison to the original algorithm LEO, the order of the operations is such that (i) neurons are not removed or merged if the entire layer is going to be folded; and (ii) special cases such as a neuron with weight vector $\boldsymbol{w}_i^l = \boldsymbol{0}$ do not need to be considered apart. For the most elaborate operations, we prove their correctness when applied to the entire layer.

**Removal of stably inactive neurons**    This operation is performed in line 25. Since the output of stably inactive neurons is always 0, we remove those neurons without affecting subsequent computations. The case in which an entire layer is stably inactive is considered separately.

**Merging of stably active neurons**    This operation is performed between lines 12 and 24. We use the following results to show how stably active neurons can be merged.

**Proposition 3.** *Let* $\mathbb{S}$ *be a set of stably active neurons in layer* $l$. *If* $r := rank(\boldsymbol{W}_\mathbb{S}^l) < |S|$ *and let* $\mathbb{T} \subset \mathbb{S}$ *be a subset of those neurons for which* $rank(\boldsymbol{W}_\mathbb{T}^l) = r$, *then the output of the neurons in* $\mathbb{S} \setminus \mathbb{T}$ *is an affine function on the output of the neurons in* $\mathbb{T}$.

**Algorithm 2** LEO++ performs exact compression of a neural network with a single operation per layer

---

1: **Input:** neural network $\left(L, \left\{(n_l, \boldsymbol{W}^l, \boldsymbol{b}^l)\right\}_{l\in\mathbb{L}}\right)$ and stable neurons $\left(\left\{(\mathbb{P}^l, \mathbb{Q}^l)\right\}_{l\in\mathbb{L}}\right)$
2: **for** $l \leftarrow 1$ **to** $L$ **do**                                                   ▷ Loops over all hidden layers
3:     **if** $|\mathbb{P}^l| = n_l$ **then**                                      ▷ Entire layer is stably inactive
4:         find output $\overline{\boldsymbol{x}}^L$ for an arbitrary input $\overline{\boldsymbol{x}}^0 \in \mathbb{X}$
5:         remove all layers except $L$, which becomes 1
6:         $\boldsymbol{W}^1 \leftarrow \boldsymbol{0}$    and    $\boldsymbol{b}^L \leftarrow \overline{\boldsymbol{x}}^L$
7:         **break**                                                   ▷ All hidden layers were collapsed
8:     **else if** $|\mathbb{P}^l| + |\mathbb{Q}^l| = n_l$ and $l < L$ **then**         ▷ Entire layer is stable, but not inactive
9:         $\boldsymbol{W}^{l+1} \leftarrow \boldsymbol{W}^{l+1}\boldsymbol{I}_{n_l}(\mathbb{Q}^l)\boldsymbol{W}^l$    and    $\boldsymbol{b}^{l+1} \leftarrow \boldsymbol{W}^{l+1}\boldsymbol{I}_{n_l}(\mathbb{Q}^l)\boldsymbol{b}^l + \boldsymbol{b}^{l+1}$
10:        remove layer $l$                                           ▷ Hidden layer was folded
11:     **else if** $l < L$ **then**
12:        $r \leftarrow \text{rank}\left(\boldsymbol{W}^l_{\mathbb{Q}^l}\right)$
13:        **if** $r < |\mathbb{Q}^l|$ and $l < L$ **then**
14:           find $\overline{\mathbb{Q}} \subset \mathbb{Q}^l$ such that $r = |\overline{\mathbb{Q}}| = \text{rank}\left(\boldsymbol{W}^l_{\overline{\mathbb{Q}}}\right)$
15:           **for every** $i \in \mathbb{Q}^l \setminus \overline{\mathbb{Q}}$ **do**
16:              find $\{\alpha^i_j\}_{j\in\overline{\mathbb{Q}}}$ such that $\boldsymbol{w}^l_i = \sum_{j\in\overline{\mathbb{Q}}} \alpha^i_j \boldsymbol{w}^l_j$
17:           **end for**
18:           **for** $k \leftarrow 1$ **to** $n_{l+1}$ **do**
19:              **for every** $j \in \overline{\mathbb{Q}}$ **do**
20:                 $w^{l+1}_{kj} \leftarrow w^{l+1}_{kj} + \sum_{i\in\mathbb{Q}^l\setminus\overline{\mathbb{Q}}} \alpha^i_j w^{l+1}_{ki}$
21:              **end for**
22:              $b^{l+1}_k \leftarrow b^{l+1}_k + \sum_{i\in\mathbb{Q}^l\setminus\overline{\mathbb{Q}}} w^{l+1}_{ki}\left(b^l_i - \sum_{j\in\overline{\mathbb{Q}}} \alpha^i_j b^l_j\right)$
23:           **end for**
24:           remove from layer $l$ every neuron $i \in \mathbb{Q}^l \setminus \overline{\mathbb{Q}}$
25:           remove from layer $l$ every neuron $i \in \mathbb{P}^l$
26:        **end if**
27:     **end if**
28: **end for**

---

*Proof.* For every $i \in \mathbb{S} \setminus \mathbb{T}$, there is a vector $\boldsymbol{\alpha}^i \in \mathbb{R}^r$ such that $\boldsymbol{w}^l_i = \sum_{j\in\mathbb{T}} \alpha^i_j \boldsymbol{w}^l_j$. Since $\boldsymbol{x}^l_i = \boldsymbol{w}^l_i \cdot \boldsymbol{x}^{l-1} + b^l_i$ for every $i \in \mathbb{S}$ because all neurons in $\mathbb{S}$ are stably active, then for every $i \in \mathbb{S} \setminus \mathbb{T}$ it follows that $\boldsymbol{x}^l_i = \sum_{j\in\mathbb{T}} \alpha^i_j \boldsymbol{w}^l_j \cdot \boldsymbol{x}^{l-1} + b^l_i = \sum_{j\in\mathbb{T}} \alpha^i_j\left(\boldsymbol{w}^l_j \cdot \boldsymbol{x}^{l-1} + b^l_j\right) + \left(b^l_i - \sum_{j\in\mathbb{T}} \alpha^i_j b^l_j\right) = \sum_{j\in\mathbb{T}} \alpha^i_j x^l_j + \left(b^l_i - \sum_{j\in\mathbb{T}} \alpha^i_j b^l_j\right).$ $\qquad\square$

**Corollary 4.** *If $\mathbb{S}$, $\mathbb{T}$, and $l$ are such as in Proposition 3, then the pre-activation output of the neurons in layer $l + 1$ is an affine function on the outputs of all neurons from layer $l$ with exception of the neurons in $\mathbb{T}$.*

*Proof.* Let $\mathbb{U} := \{1, \ldots, n_l\} \setminus \mathbb{S}$. The pre-activation output of every neuron $i$ in layer $l + 1$ is given by $y^{l+1}_i = \sum\limits_{j\in\mathbb{U}\cup\mathbb{S}} w^{l+1}_{ij} x^l_j + b^{l+1}_i = \sum\limits_{j\in\mathbb{U}\cup\mathbb{T}} w^{l+1}_{ij} x^l_j + \sum\limits_{j\in\mathbb{S}\setminus\mathbb{T}} w^{l+1}_{ij}\left(\sum\limits_{k\in\mathbb{T}} \alpha^j_k x^l_k + \left(b^l_j - \sum\limits_{k\in\mathbb{T}} \alpha^j_k b^l_k\right)\right) +$

$b^{l+1}_i = \sum\limits_{j\in\mathbb{U}} w^{l+1}_{ij} x^l_j + \sum\limits_{j\in\mathbb{T}}\left(w^{l+1}_{ij} + \sum\limits_{k\in\mathbb{S}\setminus\mathbb{T}} \alpha^k_j w^{l+1}_i k\right) x^l_j + \left(b^{l+1}_i + \sum\limits_{j\in\mathbb{S}\setminus\mathbb{T}} w^{l+1}_{ij}\left(b^l_j - \sum\limits_{k\in\mathbb{T}} \alpha^j_k b^l_k\right)\right).$
$\qquad\square$

In Algorithm 2, we use relationships implied by the proof of Corollary 4 with $\mathbb{S} = \mathbb{Q}^l$ and $\mathbb{T} = \overline{\mathbb{Q}}$ to merge stably active neurons. By adjusting the biases of the neurons in the next layer as well as the weights connecting every neuron in $\overline{\mathbb{Q}}$ with the neurons in the next layer, we assign a weight of 0 to

the connections between every neuron in $\mathbb{Q}^l \setminus \overline{\mathbb{Q}}$ and the neurons in the next layer. Hence, we simply remove all neurons in $\mathbb{Q}^l \setminus \overline{\mathbb{Q}}$ after adjusting those network parameters.

The case in which an entire layer is stably active—either before any compression is applied or once stably inactive neurons are removed—is also considered separately.

**Folding of stable layers**   This operation is performed between lines 8 and 10. We use the following results to show that stable layers can be folded in a single step.

**Proposition 5.** *If all the neurons of layer $l \in \mathbb{L} \setminus \{L\}$ are stably active, then the pre-activation output of layer $l + 1$ is an affine function on the inputs of layer $l$.*

*Proof.* Since $\boldsymbol{x}^l = \boldsymbol{W}^l \boldsymbol{x}^{l-1} + \boldsymbol{b}^l$, then $\boldsymbol{y}^{l+1} = \boldsymbol{W}^{l+1} \boldsymbol{x}^l + \boldsymbol{b}^{l+1} = \boldsymbol{W}^{l+1} \boldsymbol{W}^l \boldsymbol{x}^{l-1} + (\boldsymbol{W}^{l+1} \boldsymbol{b}^l + \boldsymbol{b}^{l+1})$. $\qquad \square$

**Corollary 6.** *If all neurons of layer $l \in \mathbb{L} \setminus \{L\}$ are stable, then the pre-activation output of layer $l + 1$ is an affine function on the inputs of layer $l$.*

*Proof.* Let $\mathbb{S}$ be the set of stably active neurons in layer $l$. If $|\mathbb{S}| < n_l$, the identity $\boldsymbol{x}^l = \boldsymbol{W}^l \boldsymbol{x}^{l-1} + \boldsymbol{b}^l$ still holds if the bias and the weights of all the connections of the neurons not in $\mathbb{S}$ with the neurons in the next layer are 0. More generally, we can thus obtain an equivalent neural network if $\boldsymbol{W}^l$ and $\boldsymbol{b}^l$ are both premultiplied by $\boldsymbol{I}_{n_l}(\mathbb{S})$ since that only would change the weights and biases associated with the neurons not in $\mathbb{S}$ to 0. Hence, the identity $\boldsymbol{x}^l = \boldsymbol{I}_{n_l}(\mathbb{S}) (\boldsymbol{W}^l \boldsymbol{x}^{l-1} + \boldsymbol{b}^l)$ always holds if all neurons in layer $l$ are stable, which implies that $\boldsymbol{y}^{l+1} = \boldsymbol{W}^{l+1} \boldsymbol{I}_{n_l}(\mathbb{S}) \boldsymbol{W}^l \boldsymbol{x}^{l-1} + (\boldsymbol{W}^{l+1} \boldsymbol{I}_{n_l}(\mathbb{S}) \boldsymbol{b}^l + \boldsymbol{b}^{l+1})$. $\quad \square$

In Algorithm 2, we use relationships implied by the proof of Corollary 6 with $\mathbb{S} = \mathbb{Q}^l$ to fold stable layers. By adjusting the biases and the weights of layer $l + 1$, that layer directly uses the outputs from layer $l - 1$.

Although the steps above would apply if a layer is stably inactive, that case deserves separate consideration.

**Collapse of a network with stably inactive layers**   This operation is performed between lines 3 and 7. If layer $l \in \mathbb{L}$ are stably inactive, then $\boldsymbol{x}^l = 0$ for any input $\boldsymbol{x}^0 \in \mathbb{X}$ and thus the value of $\boldsymbol{x}^L$ is constant. Hence, we collapse layers 1 to $L - 1$ by making the output of the remaining layer constant and equal to such value of $\boldsymbol{x}^L$.

### A5.1   On the complexity of the new algorithm

While LEO++ requires solving fewer optimization problems than LEO [79], the dependence on solving a single NP-hard problem—such as MILP formulations in general—implies an exponential worst-case complexity. Nevertheless, the progress of MILP in the past decades makes it possible to solve considerably large problems with state-of-art MILP solvers. In that context, the computational experiments are a more appropriate indicator of performance improvements than complexity considerations.

## A6 Implementation details

We now provide additional experimental results evaluating our proposed method and the baseline.

**Architecture and Loss**  We implemented the fully connected architectures in PyTorch [70]. All the networks have ReLU activations but have varying number of layers and width. For the classifiers, we pass the output through a softmax layer and use negative log-likelihood loss as the loss function. For the autoencoders, we use MSE loss as the loss function.

**Datasets and Splits**  We keep the output units at $10$ and $784$ for the MNIST dataset [53] classifiers and autoencoders, respectively. We keep the output units at $10$ and $100$ for the CIFAR-10 and the CIFAR-100 dataset [48] classifiers, respectively. We use the standard train-validation data splits of each of the datasets available in PyTorch.

**Data Augmentation**  We do not do any data augmentation of training images of the MNIST dataset as in [79] for a fair comparison. We carry out the standard data augmentation of training images of the CIFAR-10 and CIFAR-100 datasets: horizontal flipping with probability $0.5$, random rotation in the range between $(-10^o, 10^o)$, random scaling in the range $(0.8, 1.2)$, random shear parallel to the x axis in the range $(-10, 10)$, and scaling the brightness, contrast, saturation and hue by a random factor in the range $(0.8, 1.2)$.

**Optimization**  Training proceeds from scratch for $120$ epochs and starts with learning rate of $0.01$, which is decayed by a factor of $0.1$ after every $50$ epochs as in [79]. We use SGD with momentum optimizer, with a momentum of $0.9$ and batch size $128$ as in [79]. Unless stated otherwise, we use $\ell_1$ regularization. We keep the weight decay at $0$ unless otherwise stated. We consider the model saved in the last epoch as our final model.

**MILP Solver**  We solve the MILP formulations using Gurobi 9.1.0 through gurobipy [32]. We calculate the value of the positive constants $M_i^l$ and $\mu_i^l$ for each neuron with an upper bound of on the values of $x_i^l$ and $\chi_i^l$ through interval arithmetic by taking element-wise maxima [17].

**Initialization**  We initialize the weights of the network with the Kaiming initialization [40] and the biases to zero with different random seeds for each training. We train every setting $5$ times, and get the stably active and inactive neurons with the proposed approach to prune the network for each run. We omit from the summaries the runs which resulted in a time out. We keep the timeout to $3$ hours.

**Hardware**  We ran the classifier experiments on a machine with Intel Core i7-4790 CPU @ 3.60 GHz processor, 32 GB of RAM, and one 4 GB Nvidia GeForce GTX 970 GPU. The autoencoder experiments were run on a machine with $40$ Intel Xeon E5-2640 CPU @ 2.40 GHz processors, 126 GB of RAM, and one 12 GB Nvidia Titan Xp GPU.

# A7 Additional experiments and results

## A7.1 MNIST Classifiers

**Relationship between Runtime and Regularization** Tab. 1 and Tab. 2 show the runtime achieved by the proposed method at different $\ell_1$ regularization on MNIST classifiers.

Table 1: **MNIST Classifiers:** Compression results with fixed width and varying depth.

| ARCH. | $\ell_1$ | ACCURACY (%) | COMPRESSION RUNTIME (S) | % REMOVED NEURONS | CONNECTIONS | TIMED OUT |
|---|---|---|---|---|---|---|
| $2 \times 100$ | 0 | $97.92 \pm 0.09$ | $3.4 \pm 0.3$ | $0 \pm 0$ | $0 \pm 0$ | 0 |
| $2 \times 100$ | 0.000025 | $97.93 \pm 0.02$ | $3.2 \pm 0.1$ | $0 \pm 0$ | $0 \pm 0$ | 0 |
| $2 \times 100$ | 0.00005 | $98.06 \pm 0.09$ | $3.5 \pm 0.3$ | $0.1 \pm 0.2$ | $0.2 \pm 0.4$ | 0 |
| $2 \times 100$ | 0.000075 | $98.13 \pm 0.09$ | $3.2 \pm 0.2$ | $1.1 \pm 0.4$ | $2 \pm 0.8$ | 0 |
| $2 \times 100$ | 0.0001 | $98.12 \pm 0.09$ | $3.5 \pm 0.1$ | $3.4 \pm 0.7$ | $6 \pm 1$ | 0 |
| $2 \times 100$ | 0.000125 | $98.01 \pm 0.09$ | $3.5 \pm 0.3$ | $9.2 \pm 0.6$ | $17 \pm 1$ | 0 |
| $2 \times 100$ | 0.00015 | $97.9 \pm 0.1$ | $3.4 \pm 0.3$ | $12 \pm 2$ | $21 \pm 4$ | 0 |
| $2 \times 100$ | 0.000175 | $97.88 \pm 0.05$ | $3.4 \pm 0.3$ | $15 \pm 3$ | $26 \pm 4$ | 0 |
| $2 \times 100$ | 0.0002 | $97.91 \pm 0.1$ | $3.5 \pm 0.4$ | $18 \pm 2$ | $31 \pm 3$ | 0 |
| $2 \times 100$ | 0.000225 | $97.8 \pm 0.1$ | $4.2 \pm 0.9$ | $18 \pm 3$ | $31 \pm 5$ | 0 |
| $2 \times 100$ | 0.00025 | $97.65 \pm 0.09$ | $4 \pm 0.5$ | $20 \pm 2$ | $34 \pm 4$ | 0 |
| $2 \times 100$ | 0.000275 | $97.69 \pm 0.09$ | $4 \pm 1$ | $22 \pm 2$ | $38 \pm 3$ | 0 |
| $2 \times 100$ | 0.0003 | $97.64 \pm 0.06$ | $3.8 \pm 0.4$ | $24 \pm 2$ | $40 \pm 4$ | 0 |
| $2 \times 100$ | 0.000325 | $97.52 \pm 0.08$ | $3.5 \pm 0.3$ | $24 \pm 3$ | $41 \pm 4$ | 0 |
| $2 \times 100$ | 0.00035 | $97.42 \pm 0.04$ | $4 \pm 1$ | $23 \pm 3$ | $39 \pm 4$ | 0 |
| $2 \times 100$ | 0.000375 | $97.3 \pm 0.2$ | $3.4 \pm 0.3$ | $24 \pm 3$ | $40 \pm 5$ | 0 |
| $2 \times 100$ | 0.0004 | $97.28 \pm 0.03$ | $4.1 \pm 0.7$ | $23 \pm 2$ | $38 \pm 3$ | 0 |
| $3 \times 100$ | 0 | $97.86 \pm 0.06$ | $3.9 \pm 0.1$ | $0 \pm 0$ | $0 \pm 0$ | 0 |
| $3 \times 100$ | 0.000025 | $98.03 \pm 0.08$ | $10 \pm 10$ | $0 \pm 0$ | $0 \pm 0$ | 0 |
| $3 \times 100$ | 0.00005 | $98.1 \pm 0.1$ | $20 \pm 10$ | $0.1 \pm 0.3$ | $0.2 \pm 0.4$ | 0 |
| $3 \times 100$ | 0.000075 | $98.12 \pm 0.07$ | $20 \pm 20$ | $1.3 \pm 0.7$ | $1.8 \pm 1$ | 0 |
| $3 \times 100$ | 0.0001 | $98.11 \pm 0.09$ | $8 \pm 8$ | $2.7 \pm 0.9$ | $4 \pm 1$ | 0 |
| $3 \times 100$ | 0.000125 | $98.09 \pm 0.1$ | $2000 \pm 4000$ | $6 \pm 1$ | $11 \pm 3$ | 0 |
| $3 \times 100$ | 0.00015 | $98.1 \pm 0.1$ | $100 \pm 100$ | $11 \pm 2$ | $20 \pm 3$ | 0 |
| $3 \times 100$ | 0.000175 | $98.1 \pm 0.1$ | $70 \pm 60$ | $12 \pm 2$ | $20 \pm 2$ | 0 |
| $3 \times 100$ | 0.0002 | $98 \pm 0.1$ | $20 \pm 20$ | $18 \pm 2$ | $30 \pm 3$ | 0 |
| $4 \times 100$ | 0 | $97.93 \pm 0.07$ | $4.2 \pm 0.2$ | $0 \pm 0$ | $0 \pm 0$ | 0 |
| $4 \times 100$ | 0.000025 | $98 \pm 0.1$ | $200 \pm 200$ | $0 \pm 0$ | $0 \pm 0$ | 0 |
| $4 \times 100$ | 0.00005 | $98.23 \pm 0.08$ | $1000 \pm 3000$ | $0.1 \pm 0.1$ | $0.1 \pm 0.2$ | 1 |
| $4 \times 100$ | 0.000075 | $98.17 \pm 0.09$ | $1000 \pm 1000$ | $1.2 \pm 0.4$ | $1.5 \pm 0.5$ | 2 |
| $4 \times 100$ | 0.0001 | $98.1 \pm 0.06$ | $3000 \pm 3000$ | $2.8 \pm 0.9$ | $4 \pm 1$ | 2 |
| $4 \times 100$ | 0.00015 | $98.1 \pm 0.2$ | $2000 \pm 1000$ | $11 \pm 2$ | $20 \pm 4$ | 2 |
| $4 \times 100$ | 0.000175 | $98.1 \pm 0.1$ | $1000 \pm 2000$ | $14 \pm 1$ | $24 \pm 3$ | 0 |
| $4 \times 100$ | 0.0002 | $98.09 \pm 0.07$ | $1000 \pm 1000$ | $17 \pm 2$ | $30 \pm 3$ | 1 |
| $5 \times 100$ | 0 | $98.06 \pm 0.03$ | $2000 \pm 3000$ | $0 \pm 0$ | $0 \pm 0$ | 1 |
| $5 \times 100$ | 0.000025 | $98.2 \pm 0.1$ | $1000 \pm 100$ | $0 \pm 0$ | $0 \pm 0$ | 3 |
| $5 \times 100$ | 0.000175 | $98.1 \pm 0.2$ | $4000 \pm 4000$ | $15.1 \pm 0.7$ | $27 \pm 2$ | 3 |
| $5 \times 100$ | 0.0002 | $98.1 \pm 0.1$ | $3000 \pm 2000$ | $18 \pm 1$ | $32 \pm 2$ | 1 |

**Runtime Comparison with SoTA** Fig. 4 shows the comparison of runtimes with the proposed method and the baseline with the strength of $\ell_1$ regularization on the MNIST classifiers. We observe that the new method presents a median gain of **81** times in speedup.

Table 2: **MNIST Classifiers:** Compression results with fixed height and varying width.

| ARCHITECTURE | $\ell_1$ | ACCURACY (%) | COMPRESSION RUNTIME (S) | % REMOVED NEURONS | CONNECTIONS | TIMED OUT |
|---|---|---|---|---|---|---|
| $2 \times 100$ | 0 | $97.92 \pm 0.09$ | $3.4 \pm 0.3$ | $0 \pm 0$ | $0 \pm 0$ | 0 |
| $2 \times 100$ | 0.000025 | $97.93 \pm 0.02$ | $3.2 \pm 0.1$ | $0 \pm 0$ | $0 \pm 0$ | 0 |
| $2 \times 100$ | 0.00005 | $98.06 \pm 0.09$ | $3.5 \pm 0.3$ | $0.1 \pm 0.2$ | $0.2 \pm 0.4$ | 0 |
| $2 \times 100$ | 0.000075 | $98.13 \pm 0.09$ | $3.2 \pm 0.2$ | $1.1 \pm 0.4$ | $2 \pm 0.8$ | 0 |
| $2 \times 100$ | 0.0001 | $98.12 \pm 0.09$ | $3.5 \pm 0.1$ | $3.4 \pm 0.7$ | $6 \pm 1$ | 0 |
| $2 \times 100$ | 0.000125 | $98.01 \pm 0.09$ | $3.5 \pm 0.3$ | $9.2 \pm 0.6$ | $17 \pm 1$ | 0 |
| $2 \times 100$ | 0.00015 | $97.9 \pm 0.1$ | $3.4 \pm 0.3$ | $12 \pm 2$ | $21 \pm 4$ | 0 |
| $2 \times 100$ | 0.000175 | $97.88 \pm 0.05$ | $3.4 \pm 0.3$ | $15 \pm 3$ | $26 \pm 4$ | 0 |
| $2 \times 100$ | 0.0002 | $97.91 \pm 0.1$ | $3.5 \pm 0.4$ | $18 \pm 2$ | $31 \pm 3$ | 0 |
| $2 \times 100$ | 0.000225 | $97.8 \pm 0.1$ | $4.2 \pm 0.9$ | $18 \pm 3$ | $31 \pm 5$ | 0 |
| $2 \times 100$ | 0.00025 | $97.65 \pm 0.09$ | $4 \pm 0.5$ | $20 \pm 2$ | $34 \pm 4$ | 0 |
| $2 \times 100$ | 0.000275 | $97.69 \pm 0.09$ | $4 \pm 1$ | $22 \pm 2$ | $38 \pm 3$ | 0 |
| $2 \times 100$ | 0.0003 | $97.64 \pm 0.06$ | $3.8 \pm 0.4$ | $24 \pm 2$ | $40 \pm 4$ | 0 |
| $2 \times 100$ | 0.000325 | $97.52 \pm 0.08$ | $3.5 \pm 0.3$ | $24 \pm 3$ | $41 \pm 4$ | 0 |
| $2 \times 100$ | 0.00035 | $97.42 \pm 0.04$ | $4 \pm 1$ | $23 \pm 3$ | $39 \pm 4$ | 0 |
| $2 \times 100$ | 0.000375 | $97.3 \pm 0.2$ | $3.4 \pm 0.3$ | $24 \pm 3$ | $40 \pm 5$ | 0 |
| $2 \times 100$ | 0.0004 | $97.28 \pm 0.03$ | $4.1 \pm 0.7$ | $23 \pm 2$ | $38 \pm 3$ | 0 |
| $2 \times 200$ | 0 | $98.03 \pm 0.05$ | $6.9 \pm 0.7$ | $0 \pm 0$ | $0 \pm 0$ | 0 |
| $2 \times 200$ | 0.000025 | $98.2 \pm 0.05$ | $7.1 \pm 0.7$ | $0 \pm 0$ | $0 \pm 0$ | 0 |
| $2 \times 200$ | 0.00005 | $98.15 \pm 0.04$ | $7.2 \pm 0.4$ | $0.1 \pm 0.1$ | $0.2 \pm 0.3$ | 0 |
| $2 \times 200$ | 0.000075 | $98.18 \pm 0.09$ | $12 \pm 9$ | $3 \pm 1$ | $6 \pm 2$ | 0 |
| $2 \times 200$ | 0.0001 | $98.16 \pm 0.07$ | $8.8 \pm 0.7$ | $11 \pm 1$ | $20 \pm 2$ | 0 |
| $2 \times 200$ | 0.000125 | $98.1 \pm 0.09$ | $14 \pm 10$ | $15 \pm 2$ | $26 \pm 3$ | 0 |
| $2 \times 200$ | 0.00015 | $98 \pm 0.02$ | $10 \pm 3$ | $18 \pm 2$ | $32 \pm 3$ | 0 |
| $2 \times 200$ | 0.000175 | $97.9 \pm 0.1$ | $9 \pm 2$ | $20 \pm 2$ | $35 \pm 3$ | 0 |
| $2 \times 200$ | 0.0002 | $97.95 \pm 0.08$ | $8 \pm 2$ | $20.8 \pm 0.6$ | $36.6 \pm 1$ | 0 |
| $2 \times 400$ | 0 | $98.1 \pm 0.1$ | $14.8 \pm 0.4$ | $0 \pm 0$ | $0 \pm 0$ | 0 |
| $2 \times 400$ | 0.000025 | $98.25 \pm 0.09$ | $14.5 \pm 0.5$ | $0 \pm 0$ | $0 \pm 0$ | 0 |
| $2 \times 400$ | 0.00005 | $98.25 \pm 0.07$ | $20 \pm 2$ | $0 \pm 0$ | $0 \pm 0$ | 0 |
| $2 \times 400$ | 0.000075 | $98.23 \pm 0.07$ | $180 \pm 80$ | $8 \pm 1$ | $16 \pm 2$ | 0 |
| $2 \times 400$ | 0.0001 | $98.1 \pm 0.09$ | $200 \pm 100$ | $14 \pm 1$ | $26 \pm 2$ | 0 |
| $2 \times 400$ | 0.000125 | $98.05 \pm 0.08$ | $50 \pm 20$ | $18 \pm 1$ | $32 \pm 2$ | 0 |
| $2 \times 400$ | 0.00015 | $98.03 \pm 0.05$ | $29 \pm 10$ | $19 \pm 2$ | $34 \pm 3$ | 0 |
| $2 \times 400$ | 0.000175 | $97.9 \pm 0.1$ | $100 \pm 100$ | $17.7 \pm 0.8$ | $32 \pm 1$ | 0 |
| $2 \times 400$ | 0.0002 | $97.87 \pm 0.1$ | $1000 \pm 1000$ | $18 \pm 1$ | $33 \pm 2$ | 0 |
| $2 \times 800$ | 0 | $98.21 \pm 0.05$ | $37.6 \pm 0.3$ | $0 \pm 0$ | $0 \pm 0$ | 0 |
| $2 \times 800$ | 0.000025 | $98.26 \pm 0.05$ | $38.2 \pm 0.4$ | $0 \pm 0$ | $0 \pm 0$ | 0 |
| $2 \times 800$ | 0.000075 | $98.23 \pm 0.03$ | $1300 \pm 800$ | $12 \pm 0.7$ | $22 \pm 1$ | 0 |
| $2 \times 800$ | 0.0001 | $98 \pm 0.1$ | $1000 \pm 1000$ | $15.9 \pm 0.9$ | $29 \pm 1$ | 0 |
| $2 \times 800$ | 0.000125 | $98.01 \pm 0.07$ | $100 \pm 100$ | $16.8 \pm 0.8$ | $31 \pm 1$ | 0 |
| $2 \times 800$ | 0.00015 | $98.07 \pm 0.06$ | $90 \pm 30$ | $17.3 \pm 0.6$ | $31 \pm 1$ | 0 |
| $2 \times 800$ | 0.000175 | $97.91 \pm 0.07$ | $50 \pm 20$ | $16.5 \pm 0.9$ | $30 \pm 2$ | 0 |
| $2 \times 800$ | 0.0002 | $97.78 \pm 0.06$ | $80 \pm 30$ | $16.7 \pm 0.6$ | $31 \pm 1$ | 0 |

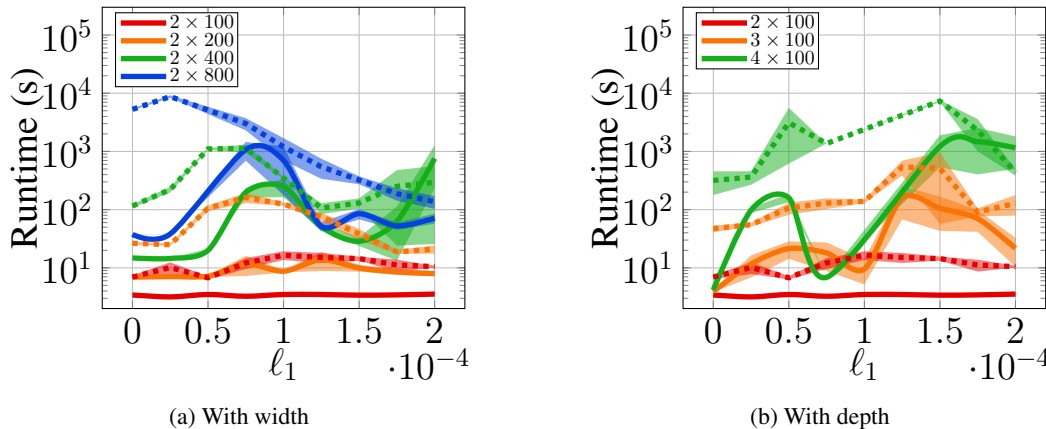

(a) With width           (b) With depth

Figure 4: **MNIST Classifiers: Comparison of runtimes** for proposed method (solid) and baseline (dashed) with the strength of regularization to identify stable neurons: (a) with increasing width (b) with increasing depth. We report the average and the standard deviation of the runtime of models with five different initialization for each regularization. Note that the y-axis is in the log scale. The median speedup is **81** times.

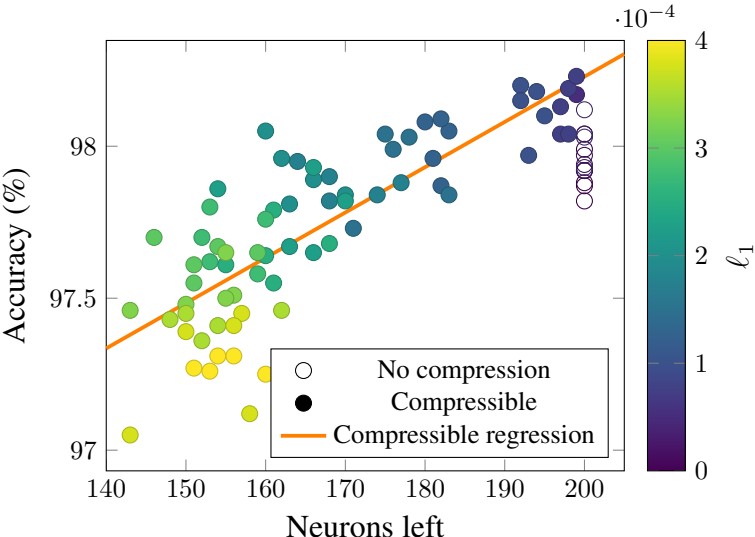

Figure 5: **Relationship between size of compressed neural network and accuracy on $2 \times 100$ MNIST classifiers.** The coefficient of determination ($R^2$) for the linear regression obtained for accuracy based on neurons left for compressible networks is 69%.

## A7.2 MNIST Autoencoders

For the autoencoders, we use the notation $n_1 \mid n_2 \mid n_3$ for the architecture of 3 hidden layers with $n_1, n_2$, and $n_3$ neurons. The output layer has the same size as the input, 784, and uses ReLU activation. Starting with the architecture $100 \mid 10 \mid 100$, we evaluated changes to the bottleneck width $n_2$ as well as to the width of the other two layers. First, we changed the bottleneck width to $n_2 = 25$ and $n_2 = 50$. Second, we changed the width of the other layers to $n_1, n_3 = 50$, $n_1, n_3 = 200$, and $n_1, n_3 = 400$ while keeping $n_2 = 10$. For each architecture, we trained and evaluated neural networks with 5 different random initialization seeds using $\ell_1 = 0$, $\ell_1 = 0.00002$, and $\ell_1 = 0.0002$.

**Relationship between Runtime and Regularization**  Tab. 3 reports the runtime to identify stable neurons and the proportion of neurons—as well as the corresponding connections—that can be removed due to stability in each case on MNIST Autoencoders.

With the largest amount of regularization, we notice that the runtimes are considerably smaller and most of the network can be removed while the loss during training only doubles in comparison to using zero or a moderate amount of regularization. In fact, the only neurons that are not stable in such case are in the first layer, whereas between 3 and 4 out of the 5 neural networks trained for each architecture have all hidden layers completely stable. By also evaluating the stability of the output layer, we identified a few cases in which the output layer is entirely stable. While we have not explicitly explored that possibility in the proposed algorithm, the implication for such case is that the neural network can be reduced to a linear function on the domain of interest. With autoencoders, we observed that this can happen when the regularization during training no more than doubles the loss, and that we can evaluate if that happens within seconds: the runtime when the stability of the output layer is tested is 1 seconds on average and never more than 25 seconds.

**Runtime Comparison with SoTA**  Fig. 6 shows the difference in runtimes between our approach and the baseline [79] for higher regularization, fixed $n_2 = 10$, and varying but equal values for $n_1$ and $n_3$ on the MNIST Autoencoders. We observe that the new method presents a median gain of **159** times in running time, which increases with the width of the non-bottleneck layers.

Table 3: **MNIST Autoencoders:** Compression results with varying architectures and levels of regularization.

| ARCHITECTURE | $\ell_1$ | LOSS | COMPRESSION RUNTIME (S) | % REMOVED NEURONS | CONNECTIONS | TIMED OUT |
|---|---|---|---|---|---|---|
| 100 \| 10 \| 100 | 0 | $0.045 \pm 0.001$ | $130 \pm 30$ | $0.1 \pm 0.1$ | $0.05 \pm 0.06$ | 0 |
| 100 \| 10 \| 100 | 0.00002 | $0.047 \pm 0.0009$ | $120 \pm 30$ | $12.7 \pm 0.6$ | $7.2 \pm 0.9$ | 0 |
| 100 \| 10 \| 100 | 0.0002 | $0.077 \pm 0.002$ | $2.73 \pm 0.05$ | $95 \pm 6$ | $90 \pm 10$ | 0 |
| 100 \| 25 \| 100 | 0 | $0.035 \pm 0.001$ | $500 \pm 300$ | $0 \pm 0$ | $0 \pm 0$ | 0 |
| 100 \| 25 \| 100 | 0.00002 | $0.047 \pm 0.001$ | $800 \pm 200$ | $14 \pm 1$ | $10 \pm 2$ | 0 |
| 100 \| 25 \| 100 | 0.0002 | $0.076 \pm 0.001$ | $2.88 \pm 0.08$ | $90 \pm 7$ | $80 \pm 20$ | 0 |
| 100 \| 50 \| 100 | 0 | $0.0311 \pm 0.0009$ | $230 \pm 20$ | $0 \pm 0$ | $0 \pm 0$ | 0 |
| 100 \| 50 \| 100 | 0.00002 | $0.0478 \pm 0.0009$ | $600 \pm 200$ | $17.4 \pm 0.9$ | $13 \pm 1$ | 0 |
| 100 \| 50 \| 100 | 0.0002 | $0.081 \pm 0.003$ | $2.96 \pm 0.04$ | $90 \pm 7$ | $80 \pm 20$ | 0 |
| 50 \| 10 \| 50 | 0 | $0.047 \pm 0.002$ | $33 \pm 4$ | $0 \pm 0$ | $0 \pm 0$ | 0 |
| 50 \| 10 \| 50 | 0.00002 | $0.051 \pm 0.002$ | $50 \pm 20$ | $14 \pm 3$ | $13 \pm 2$ | 0 |
| 50 \| 10 \| 50 | 0.0002 | $0.081 \pm 0.002$ | $1.42 \pm 0.02$ | $89 \pm 8$ | $88 \pm 8$ | 0 |
| 100 \| 10 \| 100 | 0 | $0.045 \pm 0.001$ | $130 \pm 30$ | $0.1 \pm 0.1$ | $0.05 \pm 0.06$ | 0 |
| 100 \| 10 \| 100 | 0.00002 | $0.047 \pm 0.0009$ | $120 \pm 30$ | $12.7 \pm 0.6$ | $7.2 \pm 0.9$ | 0 |
| 100 \| 10 \| 100 | 0.0002 | $0.077 \pm 0.002$ | $2.73 \pm 0.05$ | $95 \pm 6$ | $90 \pm 10$ | 0 |
| 200 \| 10 \| 200 | 0 | $0.041 \pm 0.002$ | $1000 \pm 1000$ | $0.4 \pm 0.4$ | $0.4 \pm 0.4$ | 1 |
| 200 \| 10 \| 200 | 0.00002 | $0.043 \pm 0.002$ | $700 \pm 400$ | $14 \pm 0.7$ | $7 \pm 1$ | 0 |
| 200 \| 10 \| 200 | 0.0002 | $0.076 \pm 0.002$ | $5.41 \pm 0.03$ | $95 \pm 6$ | $80 \pm 20$ | 0 |
| 400 \| 10 \| 400 | 0 | $0.04$ | $2704$ | $0$ | $0$ | 4 |
| 400 \| 10 \| 400 | 0.00002 | $0.0395 \pm 0.001$ | $1300 \pm 100$ | $15 \pm 1$ | $6 \pm 0.7$ | 0 |
| 400 \| 10 \| 400 | 0.0002 | $0.073 \pm 0.001$ | $10.5 \pm 0.2$ | $89.1 \pm 7.5$ | $13.6 \pm 59.3$ | 0 |

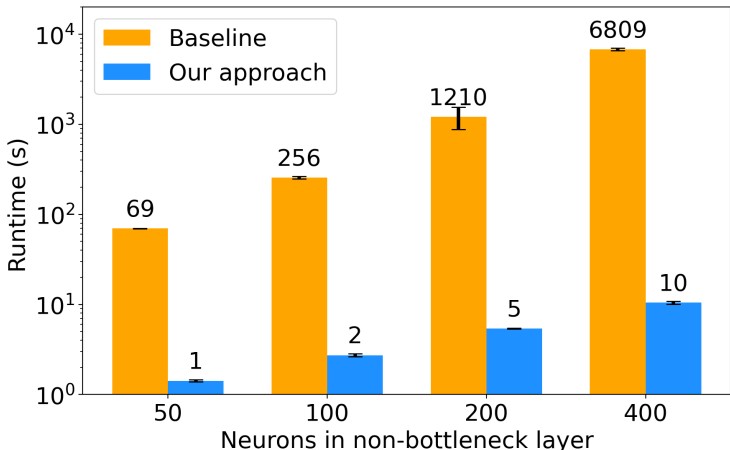

Figure 6: **MNIST Autoencoders: Comparison of runtimes** (in seconds) to identify stable neurons between the proposed approach vs. the baseline from [79] with high regularization ($\ell_1 = 0.0002$). Note that the y-axis is in the log scale. The median speedup is **159** times.

## A7.3 CIFAR-10 Classifiers

**Relationship between Runtime and Regularization** Tab. 4 and Tab. 5 show the runtime achieved by the proposed method at different $\ell_1$ regularization on the CIFAR-10 classifiers.

Table 4: **CIFAR10 Classifiers:** Compression results with fixed width and varying depth.

| | | | COMPRESSION | % REMOVED | | TIMED |
|---|---|---|---|---|---|---|
| ARCH. | $\ell_1$ | ACCURACY (%) | RUNTIME (S) | NEURONS | CONNECTIONS | OUT |
| $2 \times 100$ | 0 | $54.3 \pm 0.2$ | $13.4 \pm 0.6$ | $0 \pm 0$ | $0 \pm 0$ | 0 |
| $2 \times 100$ | 0.000025 | $53.8 \pm 0.9$ | $14 \pm 2$ | $0 \pm 0$ | $0 \pm 0$ | 0 |
| $2 \times 100$ | 0.00005 | $53.6 \pm 0.5$ | $13 \pm 3$ | $31 \pm 1$ | $56 \pm 2$ | 0 |
| $2 \times 100$ | 0.000075 | $52.7 \pm 0.6$ | $10.9 \pm 0.8$ | $34 \pm 2$ | $61 \pm 4$ | 0 |
| $2 \times 100$ | 0.0001 | $52.3 \pm 0.3$ | $11 \pm 2$ | $36 \pm 2$ | $64 \pm 2$ | 0 |
| $2 \times 100$ | 0.000125 | $51.6 \pm 0.5$ | $10.4 \pm 0.3$ | $39 \pm 3$ | $66 \pm 4$ | 0 |
| $2 \times 100$ | 0.00015 | $51 \pm 0.4$ | $11 \pm 2$ | $40 \pm 2$ | $68 \pm 3$ | 0 |
| $2 \times 100$ | 0.000175 | $50.4 \pm 0.4$ | $10.3 \pm 0.1$ | $42 \pm 3$ | $69 \pm 3$ | 0 |
| $2 \times 100$ | 0.0002 | $50.1 \pm 0.6$ | $12 \pm 2$ | $45 \pm 3$ | $71 \pm 3$ | 0 |
| $2 \times 100$ | 0.000225 | $49.6 \pm 0.4$ | $11 \pm 1$ | $45 \pm 2$ | $72 \pm 1$ | 0 |
| $2 \times 100$ | 0.00025 | $48.5 \pm 0.3$ | $10.8 \pm 0.7$ | $46 \pm 1$ | $73 \pm 2$ | 0 |
| $2 \times 100$ | 0.000275 | $48 \pm 0.4$ | $10.3 \pm 0.2$ | $47 \pm 3$ | $75 \pm 3$ | 0 |
| $2 \times 100$ | 0.0003 | $47.8 \pm 0.6$ | $10.7 \pm 0.6$ | $51 \pm 2$ | $78 \pm 2$ | 0 |
| $2 \times 100$ | 0.000325 | $47.2 \pm 0.2$ | $10.4 \pm 0.2$ | $51 \pm 3$ | $77 \pm 2$ | 0 |
| $2 \times 100$ | 0.00035 | $47.2 \pm 0.3$ | $10.5 \pm 0.5$ | $53 \pm 3$ | $79 \pm 3$ | 0 |
| $2 \times 100$ | 0.000375 | $46.8 \pm 0.4$ | $10.7 \pm 0.5$ | $54 \pm 3$ | $80 \pm 2$ | 0 |
| $2 \times 100$ | 0.0004 | $46.3 \pm 0.3$ | $10.9 \pm 0.4$ | $56 \pm 2$ | $81 \pm 2$ | 0 |
| $3 \times 100$ | 0 | $53.7 \pm 0.7$ | $13 \pm 1$ | $0 \pm 0$ | $0 \pm 0$ | 0 |
| $3 \times 100$ | 0.000025 | $54.5 \pm 0.4$ | $20 \pm 10$ | $0 \pm 0$ | $0 \pm 0$ | 0 |
| $3 \times 100$ | 0.00005 | $53.8 \pm 0.4$ | $13 \pm 2$ | $22.3 \pm 0.8$ | $32 \pm 1$ | 0 |
| $3 \times 100$ | 0.000075 | $53.3 \pm 0.6$ | $11.6 \pm 0.9$ | $23 \pm 2$ | $34 \pm 4$ | 0 |
| $3 \times 100$ | 0.0001 | $53.2 \pm 0.6$ | $20 \pm 10$ | $25 \pm 2$ | $36 \pm 3$ | 0 |
| $3 \times 100$ | 0.000125 | $52.5 \pm 0.6$ | $14 \pm 5$ | $26 \pm 2$ | $38 \pm 3$ | 0 |
| $3 \times 100$ | 0.00015 | $51.98 \pm 0.05$ | $16 \pm 6$ | $29 \pm 1$ | $43 \pm 1$ | 0 |
| $3 \times 100$ | 0.000175 | $50.8 \pm 0.6$ | $12 \pm 1$ | $32 \pm 2$ | $47 \pm 2$ | 0 |
| $3 \times 100$ | 0.0002 | $50.3 \pm 0.4$ | $15 \pm 7$ | $35 \pm 2$ | $52 \pm 3$ | 0 |
| $4 \times 100$ | 0 | $53.6 \pm 0.6$ | $20 \pm 10$ | $0 \pm 0$ | $0 \pm 0$ | 0 |
| $4 \times 100$ | 0.000025 | $53.9 \pm 0.6$ | $20 \pm 20$ | $0 \pm 0$ | $0 \pm 0$ | 0 |
| $4 \times 100$ | 0.00005 | $53.9 \pm 0.2$ | $17 \pm 8$ | $15.9 \pm 0.6$ | $20.5 \pm 0.8$ | 0 |
| $4 \times 100$ | 0.000075 | $53.7 \pm 0.3$ | $13 \pm 1$ | $17 \pm 1$ | $22 \pm 1$ | 0 |
| $4 \times 100$ | 0.0001 | $52.7 \pm 0.3$ | $60 \pm 90$ | $19.3 \pm 1$ | $25 \pm 1$ | 0 |
| $4 \times 100$ | 0.000125 | $52.4 \pm 0.6$ | $15 \pm 5$ | $21 \pm 2$ | $29 \pm 2$ | 0 |
| $4 \times 100$ | 0.00015 | $51.6 \pm 0.2$ | $600 \pm 800$ | $25 \pm 1$ | $34 \pm 2$ | 0 |
| $4 \times 100$ | 0.000175 | $50.7 \pm 0.3$ | $700 \pm 800$ | $28.5 \pm 0.9$ | $40 \pm 1$ | 1 |
| $4 \times 100$ | 0.0002 | $50.3 \pm 0.4$ | $400 \pm 400$ | $33.7 \pm 0.9$ | $49 \pm 1$ | 0 |
| $5 \times 100$ | 0 | $53 \pm 0.5$ | $14.4 \pm 0.4$ | $0 \pm 0$ | $0 \pm 0$ | 0 |
| $5 \times 100$ | 0.000025 | $53.3 \pm 0.8$ | $18 \pm 5$ | $0 \pm 0$ | $0 \pm 0$ | 0 |
| $5 \times 100$ | 0.00005 | $54 \pm 0.1$ | $30 \pm 20$ | $12.9 \pm 0.6$ | $15.7 \pm 0.7$ | 0 |
| $5 \times 100$ | 0.000075 | $53.5 \pm 0.4$ | $100 \pm 200$ | $14 \pm 0.5$ | $17.1 \pm 0.6$ | 0 |
| $5 \times 100$ | 0.0001 | $53.3 \pm 0.3$ | $11.8 \pm 0.4$ | $16 \pm 1$ | $20 \pm 1$ | 2 |
| $5 \times 100$ | 0.000125 | $51.9 \pm 0.4$ | $3000 \pm 4000$ | $14 \pm 8$ | $20 \pm 10$ | 2 |
| $5 \times 100$ | 0.00015 | $51.4$ | $1000$ | $20$ | $27$ | 4 |
| $5 \times 100$ | 0.000175 | $51.3 \pm 0.4$ | $2000 \pm 2000$ | $27.4 \pm 0.8$ | $39 \pm 1$ | 3 |
| $5 \times 100$ | 0.0002 | $50.2 \pm 0.1$ | $3000 \pm 2000$ | $31 \pm 2$ | $45 \pm 3$ | 1 |

**Runtime Comparison with SoTA** Fig. 7 shows the comparison of runtime of the proposed method and the baseline with the strength of $\ell_1$ regularization on the CIFAR-10 classifiers. We observe that the new method presents a median gain of **183** times in running time.

Table 5: **CIFAR10 Classifiers:** Compression results with fixed height and varying width.

| ARCHITECTURE | $\ell_1$ | ACCURACY (%) | COMPRESSION RUNTIME (S) | % REMOVED NEURONS | CONNECTIONS | TIMED OUT |
|---|---|---|---|---|---|---|
| $2 \times 100$ | 0 | $54.3 \pm 0.2$ | $13.4 \pm 0.6$ | $0 \pm 0$ | $0 \pm 0$ | 0 |
| $2 \times 100$ | 0.000025 | $53.8 \pm 0.9$ | $14 \pm 2$ | $0 \pm 0$ | $0 \pm 0$ | 0 |
| $2 \times 100$ | 0.00005 | $53.6 \pm 0.5$ | $13 \pm 3$ | $31 \pm 1$ | $56 \pm 2$ | 0 |
| $2 \times 100$ | 0.000075 | $52.7 \pm 0.6$ | $10.9 \pm 0.8$ | $34 \pm 2$ | $61 \pm 4$ | 0 |
| $2 \times 100$ | 0.0001 | $52.3 \pm 0.3$ | $11 \pm 2$ | $36 \pm 2$ | $64 \pm 2$ | 0 |
| $2 \times 100$ | 0.000125 | $51.6 \pm 0.5$ | $10.4 \pm 0.3$ | $39 \pm 3$ | $66 \pm 4$ | 0 |
| $2 \times 100$ | 0.00015 | $51 \pm 0.4$ | $11 \pm 2$ | $40 \pm 2$ | $68 \pm 3$ | 0 |
| $2 \times 100$ | 0.000175 | $50.4 \pm 0.4$ | $10.3 \pm 0.1$ | $42 \pm 3$ | $69 \pm 3$ | 0 |
| $2 \times 100$ | 0.0002 | $50.1 \pm 0.6$ | $12 \pm 2$ | $45 \pm 3$ | $71 \pm 3$ | 0 |
| $2 \times 100$ | 0.000225 | $49.6 \pm 0.4$ | $11 \pm 1$ | $45 \pm 2$ | $72 \pm 1$ | 0 |
| $2 \times 100$ | 0.00025 | $48.5 \pm 0.3$ | $10.8 \pm 0.7$ | $46 \pm 1$ | $73 \pm 2$ | 0 |
| $2 \times 100$ | 0.000275 | $48 \pm 0.4$ | $10.3 \pm 0.2$ | $47 \pm 3$ | $75 \pm 3$ | 0 |
| $2 \times 100$ | 0.0003 | $47.8 \pm 0.6$ | $10.7 \pm 0.6$ | $51 \pm 2$ | $78 \pm 2$ | 0 |
| $2 \times 100$ | 0.000325 | $47.2 \pm 0.2$ | $10.4 \pm 0.2$ | $51 \pm 3$ | $77 \pm 2$ | 0 |
| $2 \times 100$ | 0.00035 | $47.2 \pm 0.3$ | $10.5 \pm 0.5$ | $53 \pm 3$ | $79 \pm 3$ | 0 |
| $2 \times 100$ | 0.000375 | $46.8 \pm 0.4$ | $10.7 \pm 0.5$ | $54 \pm 3$ | $80 \pm 2$ | 0 |
| $2 \times 100$ | 0.0004 | $46.3 \pm 0.3$ | $10.9 \pm 0.4$ | $56 \pm 2$ | $81 \pm 2$ | 0 |
| $2 \times 200$ | 0 | $56.8 \pm 0.2$ | $23 \pm 2$ | $0 \pm 0$ | $0 \pm 0$ | 0 |
| $2 \times 200$ | 0.000025 | $56.8 \pm 0.6$ | $28 \pm 1$ | $0 \pm 0$ | $0 \pm 0$ | 0 |
| $2 \times 200$ | 0.00005 | $56.3 \pm 0.4$ | $30 \pm 10$ | $28 \pm 2$ | $54 \pm 3$ | 0 |
| $2 \times 200$ | 0.000075 | $55.5 \pm 0.3$ | $40 \pm 20$ | $32 \pm 2$ | $61 \pm 3$ | 0 |
| $2 \times 200$ | 0.0001 | $54.3 \pm 0.4$ | $24 \pm 6$ | $37 \pm 2$ | $68 \pm 3$ | 0 |
| $2 \times 200$ | 0.000125 | $53.3 \pm 0.3$ | $1000 \pm 2000$ | $42 \pm 1$ | $72 \pm 2$ | 0 |
| $2 \times 200$ | 0.00015 | $51.9 \pm 0.7$ | $24 \pm 4$ | $45 \pm 2$ | $75 \pm 2$ | 0 |
| $2 \times 200$ | 0.000175 | $51.2 \pm 0.4$ | $21.6 \pm 0.8$ | $49 \pm 2$ | $78 \pm 2$ | 0 |
| $2 \times 200$ | 0.0002 | $50.4 \pm 0.1$ | $23 \pm 3$ | $52 \pm 2$ | $80 \pm 1$ | 0 |
| $2 \times 400$ | 0 | $58.7 \pm 0.1$ | $48 \pm 2$ | $0 \pm 0$ | $0 \pm 0$ | 0 |
| $2 \times 400$ | 0.000025 | $59.2 \pm 0.4$ | $55 \pm 9$ | $0 \pm 0$ | $0 \pm 0$ | 0 |
| $2 \times 400$ | 0.00005 | $58.2 \pm 0.1$ | $60 \pm 30$ | $28 \pm 1$ | $54 \pm 2$ | 0 |
| $2 \times 400$ | 0.000075 | $56.1 \pm 0.2$ | $51 \pm 3$ | $37 \pm 1$ | $68 \pm 2$ | 0 |
| $2 \times 400$ | 0.0001 | $55 \pm 0.3$ | $48 \pm 4$ | $45 \pm 2$ | $75 \pm 2$ | 0 |
| $2 \times 400$ | 0.000125 | $53.5 \pm 0.2$ | $45 \pm 3$ | $48.3 \pm 0.8$ | $77.5 \pm 0.6$ | 0 |
| $2 \times 400$ | 0.00015 | $51.9 \pm 0.3$ | $50 \pm 10$ | $52 \pm 1$ | $80 \pm 2$ | 0 |
| $2 \times 400$ | 0.000175 | $50.9 \pm 0.5$ | $43 \pm 3$ | $56 \pm 2$ | $83 \pm 1$ | 0 |
| $2 \times 400$ | 0.0002 | $50.3 \pm 0.3$ | $45 \pm 3$ | $58 \pm 3$ | $83 \pm 2$ | 0 |
| $2 \times 800$ | 0 | $60.3 \pm 0.2$ | $125 \pm 7$ | $0 \pm 0$ | $0 \pm 0$ | 0 |
| $2 \times 800$ | 0.000025 | $60.3 \pm 0.2$ | $190 \pm 80$ | $0 \pm 0$ | $0 \pm 0$ | 0 |
| $2 \times 800$ | 0.00005 | $58.3 \pm 0.2$ | $240 \pm 90$ | $23 \pm 6$ | $50 \pm 10$ | 0 |
| $2 \times 800$ | 0.000075 | $56.3 \pm 0.3$ | $150 \pm 50$ | $37 \pm 9$ | $60 \pm 10$ | 0 |
| $2 \times 800$ | 0.0001 | $54.6 \pm 0.2$ | $108 \pm 9$ | $40 \pm 10$ | $70 \pm 10$ | 0 |
| $2 \times 800$ | 0.000125 | $53.2 \pm 0.5$ | $130 \pm 30$ | $50 \pm 2$ | $76 \pm 2$ | 0 |
| $2 \times 800$ | 0.00015 | $51.8 \pm 0.3$ | $110 \pm 10$ | $52.5 \pm 0.8$ | $78.2 \pm 0.7$ | 0 |
| $2 \times 800$ | 0.000175 | $50.6 \pm 0.4$ | $99 \pm 6$ | $53 \pm 1$ | $78.7 \pm 1$ | 0 |
| $2 \times 800$ | 0.0002 | $50.3 \pm 0.2$ | $98 \pm 6$ | $54 \pm 1$ | $79 \pm 1$ | 0 |

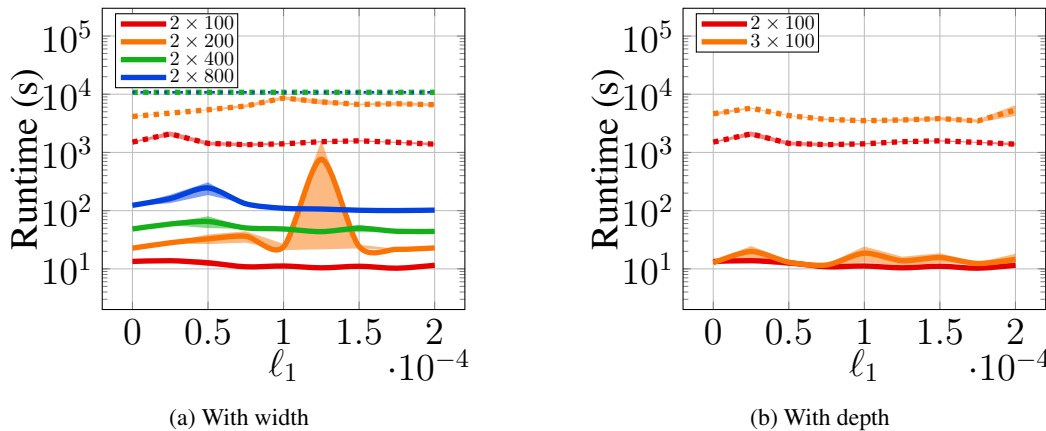

(a) With width

(b) With depth

Figure 7: **CIFAR-10 Classifiers: Comparison of runtimes** for proposed method (solid) and baseline (dashed) with the strength of regularization to identify stable neurons: (a) with increasing width (b) with increasing depth. We report the average and the standard deviation of the runtime of models with five different initialization for each regularization. Note that the y-axis is in the log scale. The median speedup is **183** times.

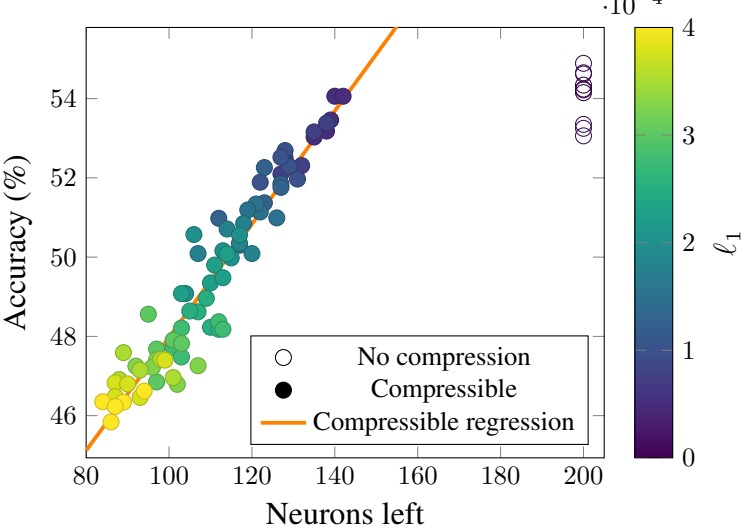

Figure 8: **Relationship between size of compressed neural network and accuracy on** $2 \times 100$ **CIFAR-10 classifiers.** The coefficient of determination ($R^2$) for the linear regression obtained for accuracy based on neurons left for compressible networks is 91%.

## A7.4 CIFAR-100 Classifiers

**Relationship between Runtime and Regularization**   Tab. 6 and Tab. 7 show the runtime achieved by the proposed method at different $\ell_1$ regularization on the CIFAR-100 classifiers.

Table 6: **CIFAR100 Classifiers:** Compression results with fixed width and varying depth.

| Arch. | $\ell_1$ | Accuracy (%) | Compression Runtime (s) | % Removed Neurons | Connections | Timed Out |
|---|---|---|---|---|---|---|
| $2 \times 100$ | 0 | $25.2 \pm 0.2$ | $13 \pm 1$ | $0 \pm 0$ | $0 \pm 0$ | 0 |
| $2 \times 100$ | 0.000025 | $24.8 \pm 0.4$ | $13 \pm 2$ | $0 \pm 0$ | $0 \pm 0$ | 1 |
| $2 \times 100$ | 0.00005 | $24 \pm 0.7$ | $11.4 \pm 0.4$ | $36 \pm 4$ | $36 \pm 4$ | 1 |
| $2 \times 100$ | 0.000075 | 23.7 | 10.4 | 42 | 42 | 4 |
| $2 \times 100$ | 0.0001 | $23.4 \pm 0.6$ | $10.3 \pm 0.1$ | $42 \pm 1$ | $43 \pm 1$ | 1 |
| $2 \times 100$ | 0.000125 | $22 \pm 2$ | $10.453 \pm 0.004$ | $48 \pm 1$ | $48 \pm 2$ | 3 |
| $2 \times 100$ | 0.00015 | $22.4 \pm 0.6$ | $10.8 \pm 1$ | $48 \pm 3$ | $48 \pm 3$ | 1 |
| $2 \times 100$ | 0.000175 | $21.5 \pm 0.5$ | $10.8 \pm 0.3$ | $47.9 \pm 0.2$ | $48.7 \pm 0.4$ | 1 |
| $2 \times 100$ | 0.0002 | $21 \pm 1$ | $10.6 \pm 0.3$ | $51 \pm 2$ | $52 \pm 2$ | 0 |
| $2 \times 100$ | 0.000225 | $21.2 \pm 0.4$ | $11 \pm 0.7$ | $51 \pm 2$ | $52 \pm 2$ | 0 |
| $2 \times 100$ | 0.00025 | $21 \pm 2$ | $10.6 \pm 0.5$ | $50 \pm 3$ | $52 \pm 4$ | 0 |
| $2 \times 100$ | 0.000275 | $20.7 \pm 0.8$ | $10.4 \pm 0.1$ | $52 \pm 2$ | $54 \pm 2$ | 0 |
| $2 \times 100$ | 0.0003 | $19 \pm 1$ | $10.6 \pm 0.2$ | $53 \pm 2$ | $55 \pm 2$ | 0 |
| $2 \times 100$ | 0.000325 | $19 \pm 1$ | $10.7 \pm 0.7$ | $53 \pm 4$ | $55 \pm 4$ | 0 |
| $2 \times 100$ | 0.00035 | $19.2 \pm 0.9$ | $11 \pm 1$ | $53 \pm 2$ | $55 \pm 1$ | 0 |
| $2 \times 100$ | 0.000375 | $19.4 \pm 0.5$ | $10.5 \pm 0.4$ | $54 \pm 2$ | $56 \pm 2$ | 0 |
| $2 \times 100$ | 0.0004 | $19 \pm 0.5$ | $10.5 \pm 0.3$ | $53 \pm 3$ | $56 \pm 3$ | 0 |
| $3 \times 100$ | 0 | $24.9 \pm 0.4$ | $16 \pm 3$ | $0 \pm 0$ | $0 \pm 0$ | 0 |
| $3 \times 100$ | 0.000025 | $25.1 \pm 0.4$ | $17 \pm 2$ | $0 \pm 0$ | $0 \pm 0$ | 2 |
| $3 \times 100$ | 0.00005 | $25.4 \pm 0.6$ | $20 \pm 10$ | $22 \pm 2$ | $22 \pm 2$ | 2 |
| $3 \times 100$ | 0.000075 | $24 \pm 1$ | $13 \pm 3$ | $28 \pm 2$ | $28 \pm 2$ | 1 |
| $3 \times 100$ | 0.0001 | $24 \pm 1$ | $11.3 \pm 0.4$ | $30 \pm 0.9$ | $30.4 \pm 1$ | 1 |
| $3 \times 100$ | 0.000125 | $24 \pm 1$ | $12 \pm 1$ | $31 \pm 1$ | $32.4 \pm 0.9$ | 1 |
| $3 \times 100$ | 0.00015 | $23.1 \pm 0.5$ | $50 \pm 80$ | $34 \pm 1$ | $37 \pm 1$ | 0 |
| $3 \times 100$ | 0.000175 | $22 \pm 1$ | $10.7 \pm 0.4$ | $36 \pm 2$ | $38 \pm 3$ | 0 |
| $3 \times 100$ | 0.0002 | $22.4 \pm 0.6$ | $12 \pm 1$ | $39 \pm 2$ | $44 \pm 3$ | 0 |
| $4 \times 100$ | 0 | $24.7 \pm 0.5$ | $30 \pm 20$ | $0 \pm 0$ | $0 \pm 0$ | 0 |
| $4 \times 100$ | 0.000025 | $25 \pm 0.7$ | $16 \pm 4$ | $0 \pm 0$ | $0 \pm 0$ | 1 |
| $4 \times 100$ | 0.00005 | $24.8 \pm 0.8$ | $2000 \pm 3000$ | $18 \pm 1$ | $18 \pm 1$ | 1 |
| $4 \times 100$ | 0.000075 | $25.1 \pm 0.5$ | $12 \pm 1$ | $20 \pm 1$ | $20 \pm 1$ | 1 |
| $4 \times 100$ | 0.0001 | $24.8 \pm 0.2$ | $12 \pm 2$ | $22 \pm 2$ | $22 \pm 2$ | 2 |
| $4 \times 100$ | 0.000125 | $23.9 \pm 0.4$ | $11.8 \pm 0.5$ | $23.9 \pm 0.4$ | $25 \pm 0.7$ | 2 |
| $4 \times 100$ | 0.00015 | $23 \pm 1$ | $50 \pm 70$ | $28 \pm 2$ | $31 \pm 3$ | 1 |
| $4 \times 100$ | 0.000175 | $22 \pm 2$ | $50 \pm 60$ | $31 \pm 3$ | $36 \pm 4$ | 0 |
| $4 \times 100$ | 0.0002 | $22 \pm 1$ | $100 \pm 200$ | $34 \pm 2$ | $41 \pm 2$ | 0 |
| $5 \times 100$ | 0 | $24.2 \pm 0.5$ | $18 \pm 4$ | $0 \pm 0$ | $0 \pm 0$ | 0 |
| $5 \times 100$ | 0.000025 | $24.6 \pm 0.4$ | $100 \pm 200$ | $0 \pm 0$ | $0 \pm 0$ | 0 |
| $5 \times 100$ | 0.00005 | $25.4 \pm 0.1$ | $40 \pm 30$ | $12.9 \pm 0.7$ | $12.9 \pm 0.7$ | 3 |
| $5 \times 100$ | 0.000075 | $24.6 \pm 0.2$ | $14.1 \pm 0.4$ | $16.4 \pm 0.3$ | $16.6 \pm 0.3$ | 2 |
| $5 \times 100$ | 0.0001 | $24 \pm 1$ | $1000 \pm 2000$ | $18 \pm 1$ | $19 \pm 2$ | 1 |
| $5 \times 100$ | 0.000125 | $24.3 \pm 0.2$ | $200 \pm 300$ | $19 \pm 1$ | $20 \pm 1$ | 2 |
| $5 \times 100$ | 0.00015 | $23.6 \pm 0.5$ | $30 \pm 20$ | $22.2 \pm 1$ | $26 \pm 2$ | 0 |
| $5 \times 100$ | 0.000175 | $22 \pm 1$ | $1000 \pm 1000$ | $26.5 \pm 0.5$ | $32.4 \pm 0.7$ | 0 |
| $5 \times 100$ | 0.0002 | $22 \pm 1$ | $1000 \pm 2000$ | $31 \pm 1$ | $39 \pm 1$ | 1 |

**Runtime Comparison with SoTA**   Fig. 9 shows the comparison of runtime of the proposed method and the baseline with the strength of $\ell_1$ regularization on the CIFAR-100 classifiers. We observe that the new method presents a median gain of **137** times in performance.

Table 7: **CIFAR100 Classifiers:** Compression results with fixed height and varying width.

| Architecture | $\ell_1$ | Accuracy (%) | Compression Runtime (s) | % Removed Neurons | Connections | Timed Out |
|---|---|---|---|---|---|---|
| $2 \times 100$ | 0 | $25.2 \pm 0.2$ | $13 \pm 1$ | $0 \pm 0$ | $0 \pm 0$ | 0 |
| $2 \times 100$ | 0.000025 | $24.8 \pm 0.4$ | $13 \pm 2$ | $0 \pm 0$ | $0 \pm 0$ | 1 |
| $2 \times 100$ | 0.00005 | $24 \pm 0.7$ | $11.4 \pm 0.4$ | $36 \pm 4$ | $36 \pm 4$ | 1 |
| $2 \times 100$ | 0.000075 | 23.7 | 10.4 | 42 | 42 | 4 |
| $2 \times 100$ | 0.0001 | $23.4 \pm 0.6$ | $10.3 \pm 0.1$ | $42 \pm 1$ | $43 \pm 1$ | 1 |
| $2 \times 100$ | 0.000125 | $22 \pm 2$ | $10.453 \pm 0.004$ | $48 \pm 1$ | $48 \pm 2$ | 3 |
| $2 \times 100$ | 0.00015 | $22.4 \pm 0.6$ | $10.8 \pm 1$ | $48 \pm 3$ | $48 \pm 3$ | 1 |
| $2 \times 100$ | 0.000175 | $21.5 \pm 0.5$ | $10.8 \pm 0.3$ | $47.9 \pm 0.2$ | $48.7 \pm 0.4$ | 1 |
| $2 \times 100$ | 0.0002 | $21 \pm 1$ | $10.6 \pm 0.3$ | $51 \pm 2$ | $52 \pm 2$ | 0 |
| $2 \times 100$ | 0.000225 | $21.2 \pm 0.4$ | $11 \pm 0.7$ | $51 \pm 2$ | $52 \pm 2$ | 0 |
| $2 \times 100$ | 0.00025 | $21 \pm 2$ | $10.6 \pm 0.5$ | $50 \pm 3$ | $52 \pm 4$ | 0 |
| $2 \times 100$ | 0.000275 | $20.7 \pm 0.8$ | $10.4 \pm 0.1$ | $52 \pm 2$ | $54 \pm 2$ | 0 |
| $2 \times 100$ | 0.0003 | $19 \pm 1$ | $10.6 \pm 0.2$ | $53 \pm 2$ | $55 \pm 2$ | 0 |
| $2 \times 100$ | 0.000325 | $19 \pm 1$ | $10.7 \pm 0.7$ | $53 \pm 4$ | $55 \pm 4$ | 0 |
| $2 \times 100$ | 0.00035 | $19.2 \pm 0.9$ | $11 \pm 1$ | $53 \pm 2$ | $55 \pm 1$ | 0 |
| $2 \times 100$ | 0.000375 | $19.4 \pm 0.5$ | $10.5 \pm 0.4$ | $54 \pm 2$ | $56 \pm 2$ | 0 |
| $2 \times 100$ | 0.0004 | $19 \pm 0.5$ | $10.5 \pm 0.3$ | $53 \pm 3$ | $56 \pm 3$ | 0 |
| $2 \times 200$ | 0 | $28.2 \pm 0.3$ | $25 \pm 3$ | $0 \pm 0$ | $0 \pm 0$ | 0 |
| $2 \times 200$ | 0.000025 | 28.5 | 29.4 | 0 | 0 | 4 |
| $2 \times 200$ | 0.00005 | $28.1 \pm 0.4$ | $27 \pm 7$ | $31 \pm 2$ | $42 \pm 3$ | 0 |
| $2 \times 200$ | 0.000075 | $27.6 \pm 0.3$ | $40 \pm 10$ | $36 \pm 1$ | $48 \pm 1$ | 0 |
| $2 \times 200$ | 0.0001 | $26.9 \pm 0.3$ | $27 \pm 9$ | $40 \pm 1$ | $52 \pm 1$ | 0 |
| $2 \times 200$ | 0.000125 | $26.1 \pm 0.3$ | $20.8 \pm 0.5$ | $44 \pm 2$ | $57 \pm 2$ | 0 |
| $2 \times 200$ | 0.00015 | $25.7 \pm 0.2$ | $21 \pm 1$ | $46 \pm 2$ | $58 \pm 2$ | 0 |
| $2 \times 200$ | 0.000175 | $25 \pm 0.3$ | $21.1 \pm 0.8$ | $48 \pm 2$ | $60 \pm 2$ | 0 |
| $2 \times 200$ | 0.0002 | $24.2 \pm 0.4$ | $21.2 \pm 0.6$ | $49.1 \pm 0.9$ | $61.6 \pm 0.9$ | 1 |
| $2 \times 400$ | 0 | $30.2 \pm 0.2$ | $46.2 \pm 0.8$ | $0 \pm 0$ | $0 \pm 0$ | 1 |
| $2 \times 400$ | 0.000025 | $30.71 \pm 0.04$ | $51 \pm 7$ | $0 \pm 0$ | $0 \pm 0$ | 2 |
| $2 \times 400$ | 0.00005 | $30.2 \pm 0.3$ | $60 \pm 10$ | $26.5 \pm 0.8$ | $42 \pm 1$ | 1 |
| $2 \times 400$ | 0.000075 | $29.13 \pm 0.09$ | $49 \pm 5$ | $33 \pm 2$ | $51 \pm 3$ | 2 |
| $2 \times 400$ | 0.0001 | $28 \pm 0.4$ | $51 \pm 7$ | $38.3 \pm 0.7$ | $56.8 \pm 0.9$ | 1 |
| $2 \times 400$ | 0.000125 | $26.8 \pm 0.4$ | $44 \pm 1$ | $43 \pm 1$ | $62 \pm 2$ | 1 |
| $2 \times 400$ | 0.00015 | $25.9 \pm 0.3$ | $47 \pm 3$ | $45 \pm 2$ | $64 \pm 2$ | 1 |
| $2 \times 400$ | 0.000175 | $25 \pm 0.2$ | $44 \pm 3$ | $47 \pm 1$ | $66 \pm 2$ | 0 |
| $2 \times 400$ | 0.0002 | $24.2 \pm 0.1$ | $44 \pm 2$ | $48 \pm 2$ | $66 \pm 2$ | 0 |
| $2 \times 800$ | 0 | $31.32 \pm 0.09$ | $100 \pm 20$ | $0 \pm 0$ | $0 \pm 0$ | 2 |
| $2 \times 800$ | 0.00005 | $30.9 \pm 0.3$ | $300 \pm 100$ | $21.4 \pm 0.8$ | $38 \pm 1$ | 0 |
| $2 \times 800$ | 0.000075 | $29.4 \pm 0.2$ | $200 \pm 100$ | $32.5 \pm 0.5$ | $52.2 \pm 0.8$ | 0 |
| $2 \times 800$ | 0.0001 | $27.8 \pm 0.3$ | $97 \pm 5$ | $39.1 \pm 0.7$ | $60 \pm 0.8$ | 0 |
| $2 \times 800$ | 0.000125 | $26.7 \pm 0.2$ | $2000 \pm 4000$ | $41 \pm 1$ | $62 \pm 1$ | 0 |
| $2 \times 800$ | 0.00015 | $25.8 \pm 0.2$ | $98 \pm 5$ | $42 \pm 1$ | $64 \pm 2$ | 0 |
| $2 \times 800$ | 0.000175 | $24.6 \pm 0.2$ | $200 \pm 100$ | $44 \pm 2$ | $65 \pm 2$ | 0 |
| $2 \times 800$ | 0.0002 | $23.6 \pm 0.5$ | $110 \pm 10$ | $44.4 \pm 1$ | $66 \pm 1$ | 0 |

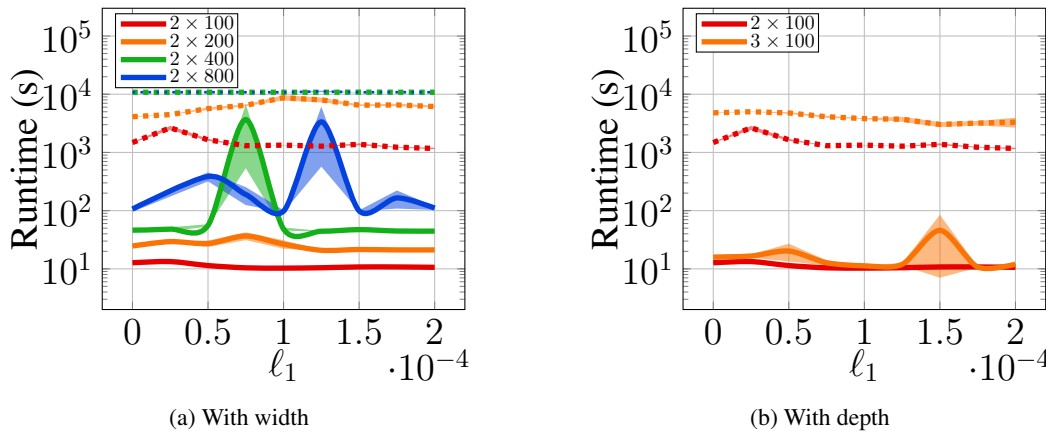

|  |  |
|---|---|
| (a) With width | (b) With depth |

Figure 9: **CIFAR-100 Classifiers: Comparison of runtimes** for proposed method (solid) and baseline (dashed) with the strength of regularization to identify stable neurons: (a) with increasing width (b) with increasing depth. We report the average and the standard deviation of the runtime of models with five different initialization for each regularization. Note that the y-axis is in the log scale. The median speedup is **137** times.

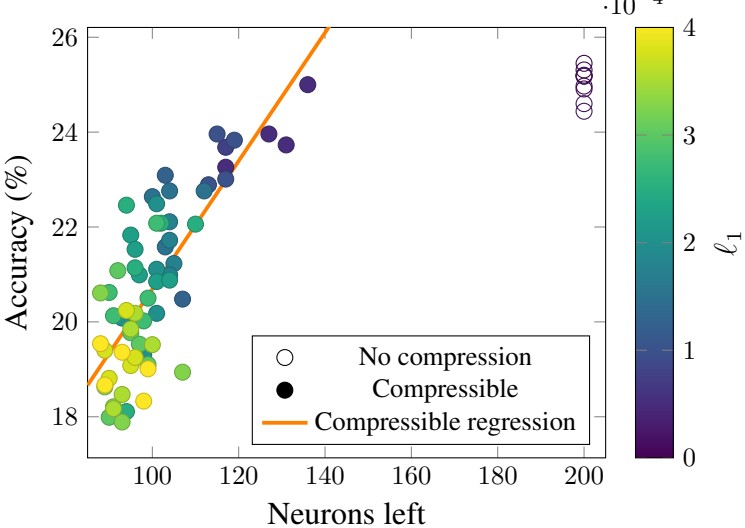

Figure 10: **Relationship between size of compressed neural network and accuracy on** $2 \times 100$ **CIFAR-100 classifiers.** The coefficient of determination ($R^2$) for the linear regression obtained for accuracy based on neurons left for compressible networks is 61%.

**A7.5    Extensions to CNNs: CIFAR-10 LeNet Classifiers**

We also test our approach with the LeNet [53] architecture on CIFAR-10 using the preprocessing step as a predictor of neuron stability to make it more scalable. We note that in this case we would only use our method as a sparsification technique to mask stably inactive zeros due to parameter sharing.

When no regularization is used and the test accuracy on CIFAR-10 is around $68.7\%$ before pruning, we find that an average of $10.98\%$ of the stably inactive neurons can be masked as $0$. With an $\ell_1$ regularization of $0.000175$, test accuracy on CIFAR-10 is around $70.02\%$ before pruning while an average of $11.86\%$ of the stably inactive neurons can be masked as $0$. In comparison to the case of MLPs, we observe more variability on the number of stable neurons across networks trained with the same amount of regularization, which we believe is due to weight sharing. Similar to the case of MLPs, the proportion of neurons that are stable in the training set but not stable in the test set is relatively small: $1.06\%$ on average. Moreover, we observe that pruning those extra neurons has a zero net effect on accuracy for regularization values in the interval $[0, 0.0003]$.

On a final note, we emphasize that masking $10\%$ of the neurons is more strict than masking $10\%$ of the parameters as done for lossy compression. Furthermore, masking $10\%$ of the neurons does not prevent someone from sparsifying the CNN even further: our method merely identifies a set of neurons—and corresponding parameters—that can be ignored for not being relevant. We believe that our method could be used in conjunction with conventional sparsification techniques in order to decompose the pruning operations of those into a lossless and a lossy component.

## A8    Extensions to Data and Batch Normalization

Normalization layers, specially Batch Normalization [45], are present in almost every modern neural network [40]. We now show how to extend our approach to these layers.

**Data Normalization**    Data normalization transforms the input $x$ as

$$\text{Norm}(x) = \frac{x - \mu}{\sigma}, \tag{20}$$

where $(\mu, \sigma)$ correspond to the mean and standard deviation of the data, respectively.

Since, we assume the image pixels to lie in the range $[0, 1]$, the data normalization layer brings the image pixels in the range $\left[-\frac{\mu}{\sigma}, \frac{1-\mu}{\sigma}\right]$. Thus, we incorporate data normalization in our approach by adjusting the input bounds using the mean and standard deviation parameters. Hence, we replace the constraint $x \in [0, 1]$ with the new constraint $x \in \left[-\frac{\mu}{\sigma}, \frac{1-\mu}{\sigma}\right]$.

**Batch Normalization**    Batch Normalization (BN) [45] corresponds to applying the affine transformation to the input $x$ as

$$\text{BN}(x) = \gamma \left( \frac{x - \mu_{\mathcal{B}}}{\sqrt{\sigma_{\mathcal{B}}^2 + \epsilon}} \right) + \beta, \tag{21}$$

where $(\mu_{\mathcal{B}}, \sigma_{\mathcal{B}}^2)$ are the mean and variance (the mini-batch statistics) of the data, $(\gamma, \beta)$ are the trainable parameters, and $\epsilon$ is a small constant to avoid division by zero.

For lossless compression, we run the MILP solver after the training of the neural network completes. Thus, BN mini-batch statistics are frozen (do not update) while running MILP, and BN only serves to scale the layer input. If the layer input before the BN layer is in the range $[h_{min}, h_{max}]$, the BN layer brings these input in the range $\left[ \gamma \left( \frac{h_{min} - \mu_{\mathcal{B}}}{\sqrt{\sigma_{\mathcal{B}}^2 + \epsilon}} \right) + \beta, \gamma \left( \frac{h_{max} - \mu_{\mathcal{B}}}{\sqrt{\sigma_{\mathcal{B}}^2 + \epsilon}} \right) + \beta \right]$. Thus, BN does not introduce any extra constraint for the MILP formulation.

We end this discussion on a final note. Although BN in inference does an affine transform of the input, BN in inference is different from the fully connected layer. BN in inference transforms the inputs individually without taking contributions from other inputs into account. On the other hand, a fully connected layer does an affine transform while taking the contributions of all inputs into account.