# OpenReview forum: "Scaling Up Exact Neural Network Compression by ReLU Stability"
_NeurIPS.cc/2021/Conference — NeurIPS 2021 Poster_

### Official Review · Reviewer_xWxW · 2021-07-14

**Rating:** 5
**Confidence:** 3

**Summary:**

This work proposes a modified Mixed-integer linear programming (MILP) to compress a feed-forward deep neural network. The proposed algorithm seems to be work if some of the network neurons are stable. The authors evaluated the algorithm with several small datasets (MNIST, CIFAR-10, and CIFAR-100).

**Limitations And Societal Impact:**

-

**Main Review:**

1.	Originality: This work is an expanded work of the previously developed method, but the authors expanded it well with some interesting techniques.

2.	Quality: The submission is technically sound overall, but some minor improvements can be expected in the experimental results.

3.	Clarity: The submission is well written and very clear.

4.	Significance: The significance of the work is moderate. Despite the authors can attain the results about 10 times faster than the baseline, the proposed algorithm still not applicable on large networks.

- Reviewer thinks the paper can be enhanced if the authors apply the algorithm to a small CNN. The CIFAR10 and CIFAR100 datasets usually hard to be achieved high accuracy on FFDNN. Fortunately, convolution can be lowered to matrix multiplication [1], the proposed algorithm can be easily expanded to CNN. Exploring this further would be helpful to the optimization community.

[1] cuDNN: Efficient Primitives for Deep Learning

- The authors mentioned the advantages of exact compression in line 33 as "there is no risk of disproportionately affecting some inputs and others". Please provide a concrete example of this.

**Time Spent Reviewing:**

3 hours

---

> ### Author Response · Authors · 2021-08-10
> **Initial response to R4 [xWxW]**
>
> We appreciate the praise and feedback on our work. Other reviewers have identified similar points, which we have addressed in our general response. We copy the most relevant parts below for your convenience.
>
> **Despite the authors can attain the results about 10 times faster than the baseline, the proposed algorithm is still not applicable on large networks**
>
> One possible simplification of our method would be to use only the preprocessing step described in Section 5.3 to determine which neurons to remove, since that step only takes a few seconds. Note that the preprocessing step is in principle intended to identify neurons that are not stable in order to avoid spending further time on them, but we can conversely assume that all the other neurons are stable at the cost of removing more than we should.
>
> By using preprocessing alone, our experiments would identify on average 31.93% potentially stable neurons in MNIST classifiers, 40.86% in CIFAR-10 classifiers, and 41.98% in CIFAR-100 classifiers. Among those neurons, only a few are actually not stable when evaluated with the test set. In relative terms, which we calculate as the number of not stable neurons with respect to the test set divided by the number of stable neurons with respect to the training set, we would have removed 1.16% more neurons that we should for MNIST,  0.60% for CIFAR-10, and 1.19% for CIFAR-100.
>
>
> **Reviewer thinks the paper can be enhanced if the authors apply the algorithm to a small CNN.**
>
> Following the suggestion of R2 (9nxn), we have tested our method with the LeNet architecture on CIFAR-10 using the preprocessing step as a predictor of neuron stability to make it more scalable. We note that in this case we would only use our method as a sparsification technique to mask stably inactive zeros due to parameter sharing.
>
> When no regularization is used and the test accuracy on CIFAR-10 is around 68.7% before pruning,  we have found that an average of 10.98% of the stably inactive neurons can be masked as 0. With an L1 regularization of 0.000175, test accuracy on CIFAR-10 is around 70.02% before pruning while an average of 11.86% of the stably inactive neurons can be masked as 0. In comparison to the case of MLPs, we observed more variability on the number of stable neurons across networks trained with the same amount of regularization, which we believe is due to weight sharing. Similar to the case of MLPs, the proportion of neurons that are stable in the training set but not stable in the test set is relatively small: 1.06% on average. Moreover, we have observed that pruning those extra neurons has a zero net effect on accuracy for regularization values in the interval [0, 3.0e-04].
>
> On a final note, we emphasize that masking 10% of the neurons is more strict than masking 10% of the parameters as done for lossy compression. Furthermore, masking 10% of the neurons does not prevent someone from sparsifying the CNN even further: our method is merely identifying a set of neurons - and corresponding parameters - that can be ignored for not being relevant. We believe that our method could be used in conjunction with conventional sparsification techniques in order to decompose the pruning operations of those into a lossless and a lossy component.
>
>
>
> **The authors mentioned the advantages of exact compression in line 33 as "there is no risk of disproportionately affecting some inputs and others". Please provide a concrete example of this.**
>
> Regarding the clarification asked by R4 in page 1, the comment in line 34 refers to prior statement in line 27:
>
> **They may also disproportionately affect some inputs more than others [42].**
>
> We understand that this explanation is too brief, and we intend to rephrase it as follows:
>
> **They may also lead to models in which the relative accuracy for some classes is more affected than that of other classes [42].**
>
> In line 34, we will replace “some inputs” with “some classes”.

---

### Official Review · Reviewer_nANE · 2021-07-14

**Rating:** 7
**Confidence:** 4

**Summary:**

This paper proposes a method to solve the problem of exact compression of a neural network, where the network is decreased in size as much as possible,  with the result being functionally equivalent to the original network over the input domain. The proposed method identifies stably active and inactive neurons and remove a part of them from the network, while demonstrating significant speedup when compared to previous work.

**Limitations And Societal Impact:**

The authors have adequately discussed the main limitation of their work, which is the difficulty to apply it in large networks. They have also adequately discussed its possible societal impact.

**Main Review:**

This paper regards the problem of decreasing the size of a neural network as one of functional equivalence. The authors seek to create a smaller network which, when a particular input domain is considered, corresponds to the same function as the original. This is considerably different from the prevailing goal in related literature, which is to decrease the size of the network while considering only its performance according to a particular metric, over the input domain.

As in previous work performed by Serra et al. (2020), the method proposed identifies the stably active and inactive neurons of the network and removes them (the former by including some of them in that of the rest of the layer, and the latter by just removing them outright). Given the set of stably active and inactive neurons, the method to remove them which is proposed in this paper (Algorithm 1) is a variation of that present in Serra et al. The main novelty of this work is how these neurons are identified, using a single MILP per layer (presented in equations (7) - (12) as well as Algorithm 2 in the supplementary - which I would suggest including in the main paper). The proposed method is technically sound as a whole and provides demonstrable improvement when runtime is concerned.

Nevertheless, I have some concerns regarding the motivation and the limitations of this paper. As noted by the authors, their proposed method involves solving an MILP per each layer of the network, which is very expensive for large networks. My main concern is whether this expensive procedure should be considered, given that the problem of exact compression appears to be very demanding as a task, especially since regular pruning methods (which disregard whether the compression is exact or not) work faster and with small reduction in common evaluation metrics of a neural network (e.g. accuracy).

The authors highlight in the Related Work section interesting cases where exact compression might be preferred, namely classification problems on imbalanced datasets, as well as robustness. I believe that including such an experiment as a motivating example (i.e. for the first case, training a network for such a task, demonstrating failure of regular pruning techniques, and then demonstrating how much compression can be obtained by the exact compression scheme) would greatly strengthen the motivation behind the paper.

Regarding the experimental evaluation present in the paper, it sufficiently demonstrates the speedup of the proposed method, as well as how much compression can be obtained under the exact scheme, both with and without regularization. I want to mention the fact that in Figure 3 and in most results in the supplementary, it appears that if there is no L1 regularization, then no nodes are removed. This seems interesting (since it demonstrates that the exact compression appears to require regularization to work) and I think it requires further discussion.

The paper is clearly written, and I only have minor comments to make:
- As mentioned above, I would suggest including Algorithm 2 in the main paper, since it contains most of the novelty of the work (or at the very least better explain it in Section 5).
- There are a couple of typos/missing words, in lines 26 and 303.
- I would suggest adding comments in Algorithm 1, to clearly identify which part of the text corresponds to which part of the pseudocode.

Overall, this is a good submission, but it lacks a few motivating examples to demonstrate the usefulness of exact compression. With those, I believe the significance of this work would be improved.

Reference:

T. Serra, , A. Kumar, and S. Ramalingam. Lossless compression of deep neural networks. In CPAIOR, 2020.

**Post-rebuttal comment**:

After the author response, I have updated my score (see below comment for details).

**Time Spent Reviewing:**

7

---

> ### Author Response · Authors · 2021-08-10
> **Initial response to R3 [nANE]**
>
> Thank you for the careful feedback on our work. We also thank you for considering our work as good and clearly written.Some of your points were also identified by other reviewers, and thus included in our main response.  For your convenience, we copy them below.
>
> **Algorithm 2 in the supplementary - which I would suggest including in the main paper [...] I would suggest including Algorithm 2 in the main paper, since it contains most of the novelty of the work [...] I would suggest adding comments in Algorithm 1, to clearly identify which part of the text corresponds to which part of the pseudocode.**
>
> From the general response:
>
> We appreciate that R2 and R3 analyze the supplemental material in detail, and we agree with the recommendations of two reviewers, which we have combined as follows:
> - Bring Algorithm 2 from the supplemental material to the paper.
> - Move Algorithm 1 and related theoretical results in Section 5.4 to supplemental material.
> - Add comments to Algorithm 1 to make it easier to follow.
> - Use space gained in the paper to add some of the explanations and experimental results produced for the rebuttal.
>
> **their proposed method involves solving an MILP per each layer**
>
> We actually solve a single MILP formulation. However, we apply all the compression operations of a given layer just once.
>
> **Demonstrating failure of regular pruning techniques, and then demonstrating how much compression can be obtained by the exact compression scheme**
>
> We could not complete this request by August 10, but we intend to follow up on that.
>
> **it appears that if there is no L1 regularization, then no nodes are removed. This seems interesting (since it demonstrates that the exact compression appears to require regularization to work) and I think it requires further discussion.**
>
> That is one reason why we have extensively tested for varying levels of L1 regularization. We copy below the part of our general response that we believe addresses your point:
>
> When used in moderate amounts, regularization improves accuracy and very often that also coincides with exact compression being possible. We believe that the negative impression with the larger plots is due to how far we have stretched the experiments on 2x100 classifiers to show that more regularization leads to even more compression, which is actually just an extra but minor result of our experiments. That is particularly the case for the smallest architectures tested, and two of those (h and o) are among the exceptions to the rule that regularization improves accuracy and leads to exact compression.
>
> We observe regularization improving accuracy in 17 of the 21 plots in Figure 3. In 14 of those cases, we can reduce the size of these more accurate networks. Those include all architectures for MNIST (a to g), the architecture with width 400 for CIFAR-100 (r), and all the architectures with 3 hidden layers for CIFAR-10 and CIFAR-100 (j, l, n, q, s, u).
>
> In order to make that more evident, we will change the background color of every plot in the range of regularization values for which (a) the accuracy is better than with no regularization and (b) some amount of exact compression is observed. We can also make these regions more evident by not plotting the results for very large amounts of regularization, if the reviewers think that this would make the main message more clear to the readers.
>
>
> **There are a couple of typos/missing words, in lines 26 and 303.**
>
> Thank you. We will correct “that not equivalent” to “that are not equivalent” in line 26; we have reworded the paragraph containing line 303.

---

> > ### Comment · Reviewer_nANE · 2021-08-22
> > **Follow-up comments**
> >
> > Thank you very much for your very detailed responses to both my and the rest of the reviewers' comments.
> >
> > With respect to the number of MILPs solved, I'd like to thank you for clearing up my misunderstanding. Going over the paper again, I would also suggest including "for every layer $l$" in Proposition 1 as a minor edit which I think improves clarity.
> >
> > Given the above, as well as the added discussion about the role of regularization, I am willing to raise my score. Furthermore, as I previously mentioned, I believe the significance would be greatly improved, if a simple motivating example for exact compression is included (similar to the one I described in my initial review).

---

> > > ### Author Response · Authors · 2021-08-26
> > > **Motivation for exact compression**
> > >
> > > In order to motivate the exact approach in comparison to commonly used methods such as magnitude-based pruning (MP), we have performed an experiment which shows that MP fails to prune some weights that provide absolutely no benefit to the model. Those are large weights associated with stably inactive neurons.
> > >
> > > First, we identified all the connections that would be pruned by the removal of stably inactive neurons with our approach, which would also be harmless if identified and removed by MP. Second, we ranked all the connections based on the absolute value of their coefficients in order to identify at what pruning ratio those connections would have been removed by MP. We consistently found out across architectures of different sizes and levels of regularization that some of the pruned connections by our method would be found at the 99th percentile. In other words, even though they would have no impact if removed from the network, MP would only resort to removing them at very extreme levels of pruning.
> > >
> > > Furthermore, if the same pruning ratio is used with MP, on average 10% of the total number of connections (or 18% of the pruned connections) removed by our method would not be removed by MP.

---

> > > > ### Comment · Reviewer_nANE · 2021-08-27
> > > > **Follow-up comment**
> > > >
> > > > Thank you very much for presenting this motivating example for your method. This is an interesting effect, and I believe it can be further elaborated on in the final paper.
> > > >
> > > > I have updated my score accordingly.

---

### Official Review · Reviewer_9nxn · 2021-07-15

**Rating:** 7
**Confidence:** 4

**Summary:**

The paper builds upon prior work on MLPs with ReLU activations that leverage the insight that ReLUs are either linear or zero to formulate a MILP to solve the problem of _exact compression_, i.e., the process of reducing neurons in the networks while exactly preserving the output of the network for the input domain. The compression hinges on the insight that if a ReLU neuron is always activated it becomes an affine function while if it's always inactive it's simple zero and does not contribute to the output.

Their main contribution over prior work is a simple, but elegant insight on the problem formulation for the MILP that avoids solving the MILP exactly for each neuron in the network and instead solving it approximately at most N times (N = # neurons) while simplifying the problem (i.e. removing decision variables pertaining to neurons that are not stable) at each step so that only during the final call the MILP has to be solved exactly. The resulting exact compression problem thus has significantly reduced runtime compared to prior work leading to the ability to scale to larger networks (up to 2x800 or 5x100 in their experiments compared to 2x100 in prior work).

**Ethical Concerns:**

No ethical concerns

**Limitations And Societal Impact:**

Yes, appropriately addressed, although I would tone down the 2nd part about societal impact with regards to "minoritized groups in society". While I get the authors' point, I think it's too much of a stretch.

**Main Review:**

## Score

I like the paper and I think that MILP formulations are powerful tools with broad range of applications for ReLU networks including compression, robustness, verification, etc... As such, the contribution to improve upen and scale MILP-based problem formulations in ReLU networks is an important contribution to the community on its own and hopefully many other papers can benefit from the insights presented in this work.

However, I do have some concerns regarding the application to compression and the resulting compression results. My main concern is that the experimental results (e.g. Fig 3a, 3h, 3o) indicate that you can only start doing meaningful compression (meaningful in the sense that you can remove a significant amount of neurons) once you increase l1-regularization, which however at that point already hurts the test accuracy. So with this method it seems you cannot get compressed networks without loss of accuracy compared to the best performing network.


## Ways to Improve my Score

Given the interesting insights into the problem formulation and the resulting impressive performance in terms of scalability of the MILP, I believe the paper needs to significantly improve some of its discussion and presentation of the experimental results. I am listening my concerns ("Weaknesse") in more details below. If the authors can address these concerns sufficiently during the rebuttal I would be happy to increase my score.


## Strengths

* Very insightful and impactful re-formulation of MILP-based problem formulations for ReLU networks. These insights may also be useful beyond the compression community.

* Strong introduction and presentation of related work.

* Easy-to-follow and good writing style.

* Cool presentation of the main contributions in Sections 5.1-5.3. I learned something while reading the paper.


## Weaknesses

* My main issues comes down to the experimental performance, mainly around Fig. 3. It seems to me that you need a lot of l1-regularization in order to be able to prune any non-negligible number of neurons. However, at that point the test accuracy has already dropped significantly compared to the networks that are only slightly regularized or unregularized. Please consider discussing some of these aspects in more detail in the paper and try to understand (even on an intuitive level where this is coming from). Alternatively, can you find scenarios where this is not the case? Maybe relaxing the search space for the input space could be one way because assuming the entire $[0,1]$ domain for MNIST, e.g., might still be too large for what is a reasonable input?

* Even if there is not an obvious scenario where the compression is more useful without accuracy loss, the authors could be discussing other potential applications or use cases of their novel MILP-formulations in order to increase the impact of the paper and the possible applications so that other researchers can benefit from their contribution.

* Could you discuss applications to other type of networks? What about convolutional layers for example? Even simple CNNs like LeNet5 could be interesting. There should be a hopefully ease generalization of the formulations to convolutional layers and I would love to see some preliminary results or discussions in that direction

* I am not a big fan of Section 5.4. I think it's too formal for something that intuitively makes a lot of sense. I would move all the theoretical results and the pseudo-code into the appendix for people if they are interested in seeing a very formal discussion about how neuron stability can be used for compression. In the main paper, I would instead focus on a high-level summary of this section. I would then use the space to expand upon some of the additional technical details with regards to Sections 5.1-5.3, i.e., move some of the material from supplementary Sections A-E into the main paper potentially even including Algorithm 2.


## Minor Feedback

* line 26: typo: "that not equivalent"

* A few more interesting and recent references for your related work:
  * On robustness of pruning (line 65): https://arxiv.org/abs/2103.03014
  * 2nd order prune methods (line 73,74): https://arxiv.org/abs/2004.14340

* The first sentence in the abstract is somewhat mis-leading. It is really only true for ReLU networks I believe.

* Maybe mention how you would handle batch-norm and data normalization in the first layer. Might be obvious some readers but potentially not to everyone.

* Lines 121-125: Could be a little more specific what the decision variables of the MILP are? That took a while to understand and could be much streamlined if you mention that more explicitly.

* Thank you for submitting a detailed appendix and code. Very helpful and very much appreciated.

## Update after Discussion

The authors have responded to my concerns throughout the rebuttal period and clarified lingering points of confusion I raised in my initial review. Specifically, they addressed all my concerns mentioned in the "Weaknesses" section of my initial review and added additional insightful experiments. I want to emphasize that the additional results definitely should make it into the final submission if accepted.

I have raised my score from 6 to 7.

**Time Spent Reviewing:**

6

---

> ### Author Response · Authors · 2021-08-10
> **Initial response to R2 [9nxn], Part I**
>
> We appreciate the thorough review and detailed suggestions for improving the papr. We further thank you for considering our submission to be clear, well-written and comprehensive. We also deeply appreciate for considering our submission to be insightful, impactful, and useful beyond the compression community. Some of the points that you have raised were also identified by other reviewers. In such cases, we have copied parts of our general response below for your convenience.
>
> **My main concern is that the experimental results (e.g. Fig 3a, 3h, 3o) indicate that you can only start doing meaningful compression (meaningful in the sense that you can remove a significant amount of neurons) once you increase l1-regularization, which however at that point already hurts the test accuracy. [...] My main issues comes down to the experimental performance, mainly around Fig. 3. It seems to me that you need a lot of l1-regularization in order to be able to prune any non-negligible number of neurons. However, at that point the test accuracy has already dropped significantly compared to the networks that are only slightly regularized or unregularized. Please consider discussing some of these aspects in more detail in the paper and try to understand (even on an intuitive level where this is coming from).**
>
> Thank you for making your point very specific here as well. The following is a copy of the general response:
>
> When used in moderate amounts, regularization improves accuracy and very often that also coincides with exact compression being possible. We believe that the negative impression with the larger plots is due to how far we have stretched the experiments on 2x100 classifiers to show that more regularization leads to even more compression, which is actually just an extra but minor result of our experiments. That is particularly the case for the smallest architectures tested, and two of those (h and o) are among the exceptions to the rule that regularization improves accuracy and leads to exact compression.
>
> We observe regularization improving accuracy in 17 of the 21 plots in Figure 3. In 14 of those cases, we can reduce the size of these more accurate networks. Those include all architectures for MNIST (a to g), the architecture with width 400 for CIFAR-100 (r), and all the architectures with 3 hidden layers for CIFAR-10 and CIFAR-100 (j, l, n, q, s, u).
>
> In order to make that more evident, we will change the background color of every plot in the range of regularization values for which (a) the accuracy is better than with no regularization and (b) some amount of exact compression is observed. We can also make these regions more evident by not plotting the results for very large amounts of regularization, if the reviewers think that this would make the main message more clear to the readers.
>
> **Alternatively, can you find scenarios where this is not the case? Maybe relaxing the search space for the input space could be one way because assuming the entire [0,1] domain for MNIST, e.g., might still be too large for what is a reasonable input?**
>
> Your comment led to some interesting discussions this week. Likewise, the fact that you were very specific about what we could improve was very helpful. From our general response:
>
> As a preliminary investigation, we have tested the effect of bounding the sum of all the MNIST inputs to be within the minimum and maximum observed values. In particular, we have constrained the sum of all inputs to be within the interval [15, 320] instead of [0, 784].  We believed that this approach would be preferable to constraining the value of individual inputs, since that would have affected the output upon rotation and translation.
>
> Note that this constraint is equivalent to imposing a prior on the number of foreground pixels on the digits to be within a range. Along the same lines, global priors that jointly act on all the pixels have been used in computer vision papers (e.g. label costs in Delong et al. IJCV-2012) in the pre-deep learning era.
>
> With this change we have obtained a better runtime in 69.6% of the cases after excluding those in which the time limit of 10,800 seconds was reached with and without this change. By restricting the analysis to the cases in which the time limit has not been exceeded either before or after the change, the geometric mean of the runtimes goes down by 17.7%.
>
> **The authors could be discussing other potential applications or use cases of their novel MILP-formulations in order to increase the impact of the paper and the possible applications so that other researchers can benefit from their contribution**
>
> We agree. Perhaps the most immediate application would be on evaluating the robustness of neural networks, for which Tjeng et al. (2019) [citation 84 in the paper] have shown that identifying stably active and inactive neurons improves performance considerably.
>
> **Could you discuss applications to other type of networks? What about convolutional layers for example? Even simple CNNs like LeNet5 could be interesting. There should be a hopefully ease generalization of the formulations to convolutional layers and I would love to see some preliminary results or discussions in that direction**
>
> From our general response:
>
> Following the suggestion of R2 (9nxn), we have tested our method with the LeNet architecture on CIFAR-10 using the preprocessing step as a predictor of neuron stability to make it more scalable. We note that in this case we would only use our method as a sparsification technique to mask stably inactive zeros due to parameter sharing.
>
> When no regularization is used and the test accuracy on CIFAR-10 is around 68.7% before pruning,  we have found that an average of 10.98% of the stably inactive neurons can be masked as 0. With an L1 regularization of 0.000175, test accuracy on CIFAR-10 is around 70.02% before pruning while an average of 11.86% of the stably inactive neurons can be masked as 0. In comparison to the case of MLPs, we observed more variability on the number of stable neurons across networks trained with the same amount of regularization, which we believe is due to weight sharing. Similar to the case of MLPs, the proportion of neurons that are stable in the training set but not stable in the test set is relatively small: 1.06% on average. Moreover, we have observed that pruning those extra neurons has a zero net effect on accuracy for regularization values in the interval [0, 3.0e-04].
>
> On a final note, we emphasize that masking 10% of the neurons is more strict than masking 10% of the parameters as done for lossy compression. Furthermore, masking 10% of the neurons does not prevent someone from sparsifying the CNN even further: our method is merely identifying a set of neurons - and corresponding parameters - that can be ignored for not being relevant. We believe that our method could be used in conjunction with conventional sparsification techniques in order to decompose the pruning operations of those into a lossless and a lossy component.
>
>
> **I am not a big fan of Section 5.4. I think it's too formal for something that intuitively makes a lot of sense. I would move all the theoretical results and the pseudo-code into the appendix for people if they are interested in seeing a very formal discussion about how neuron stability can be used for compression. In the main paper, I would instead focus on a high-level summary of this section. I would then use the space to expand upon some of the additional technical details with regards to Sections 5.1-5.3, i.e., move some of the material from supplementary Sections A-E into the main paper potentially even including Algorithm 2.**
>
> From our general response:
>
> We appreciate that R2 and R3 analyze the supplemental material in detail, and we agree with the recommendations of two reviewers, which we have combined as follows:
> - Bring Algorithm 2 from the supplemental material to the paper.
> - Move Algorithm 1 and related theoretical results in Section 5.4 to supplemental material.
> - Add comments to Algorithm 1 to make it easier to follow.
> - Use space gained in the paper to add some of the explanations and experimental results produced for the rebuttal.
>
> **line 26: typo: "that not equivalent"**
>
> Corrected to “that are not equivalent”.
>
> **A few more interesting and recent references for your related work**
>
> Thank you for bringing these nice papers to our attention. We will definitely cite them.
>
> **The first sentence in the abstract is somewhat mis-leading. It is really only true for ReLU networks I believe.**
>
> Agreed. We will replace “neural network” with “rectifier network”.

---

> > ### Author Response · Authors · 2021-08-10
> > **Initial response to R2 [9nxn], part II**
> >
> >
> > **Maybe mention how you would handle batch-norm and data normalization in the first layer. Might be obvious to some readers but potentially not to everyone.**
> >
> > We will include the following comment in the paper:
> >
> > We can incorporate data normalization to our approach by adjusting the bounds for each input using parameters for the mean $\mu$ and standard deviation $\sigma$, hence replacing $x \in [0, 1]$ with $x \in [-\mu/\sigma, (1-\mu)/\sigma]$.
> >
> > For lossless compression, MILP is run after the training is done. Thus, Batch Normalization (BN) mini-batch statistics $(\mu, \sigma)$ are frozen (do not update) while running MILP and BN only serves to scale the layer input. Hence, we can do the same in the context of BN. With the trainable parameters $\gamma$ and $\beta$ of the BN, applying BN corresponds to applying the affine transformation BN$(x) = \gamma (x - \mu) / \sigma  + \beta$ to the layer input. If the layer input before batch normalization is in $[h_{min}, h_{max}]$, then after batch normalization it becomes $[ \gamma (h_{min} - \mu) / \sigma  + \beta,  \gamma (h_{max} - \mu) / \sigma  + \beta]$.
> >
> > Thus, BN does not introduce any extra constraint for the MILP formulation. However, BN in inference is different from the fully connected layer. BN in inference transforms the activation individually **without** taking contribution from other activations into account. On the other hand, fully connected layers do an affine transform while taking the contribution of all activations into account.
> >
> > **Lines 121-125: Could be a little more specific what the decision variables of the MILP are? That took a while to understand and could be much streamlined if you mention that more explicitly.**
> >
> > We will explicitly mention the decision variables of the MILP in the final version.
> >
> > **I would tone down the 2nd part about societal impact with regards to "minoritized groups in society". While I get the authors' point, I think it's too much of a stretch.**
> >
> > We agree. This was a worst-case analysis based on the relevant literature on fairness in compression. More on the general answer copied below:
> >
> > Regarding the correction asked by R2 in page, we propose rephasing it as follows:
> >
> > “Large models are resource-intensive for both training as well as inference. In contrast to approximate methods, our exact model compression algorithms can help deep learning practitioners to save computational time and resources without worrying about any loss in performance. That helps preventing the documented side effect of disproportionally degrading performance for some classes more than for other classes when the indicator of a successful compression is the overall performance, which could also lead to fairness issues as discussed in the relevant literature [SOURCES].”
> >
> > In addition to citing the existing reference [42], we will add the following papers:
> >
> > https://arxiv.org/abs/2009.09936
> >
> > https://arxiv.org/abs/2010.03058c

---

> > ### Comment · Reviewer_9nxn · 2021-08-17
> > **A few more clarifications**
> >
> > Thank you for providing a detailed response to all my questions. I do appreciate the authors' effort in trying to address all of my concerns in a detailed manner. I am very much inclined to raise my score at this point since my questions have been answered mostly satisfactorily. However, I would like to inquire about a few additional details at this point.
> >
> > ### Figure 3
> >
> > I am still not entirely convinced from your answer that this is the case. In at least some of the sub-figures in Figure 3 it seems that the initial sharp increase in "Nodes removed" coincides with initial drop in "Test accuracy".
> >
> > However, I do like the suggestions put forth by the authors on how to adopt the plots. Could you send me an anonymized link with these updated figures so I could take a look? (This is allowed as far as I know, [see the FAQs](https://neurips.cc/Conferences/2021/PaperInformation/NeurIPS-FAQ))
> >
> > ### Additional experiments
> >
> > Please make sure that you include the CNN results and the results with the reduced range for MNIST in your final manuscript. They are really interesting and should make it into the paper.

---

> > > ### Author Response · Authors · 2021-08-17
> > > **Anonymized link for new version of Figure 3**
> > >
> > > Following up on the reviewer's request, we have created an anonymous Google account for the sole purpose of sharing an updated version of Figure 3.
> > >
> > > Please log out of Google and then use the following link:
> > > https://drive.google.com/file/d/1ANO7N8wgP3beR5oY84wAogiHCq4On_d0/view
> > >
> > > We will also update the final manuscript with the CNN results and the results with reduced range for MNIST.

---

> > > ### Comment · Reviewer_9nxn · 2021-08-19
> > > **Great rebuttal**
> > >
> > > Dear Authors,
> > >
> > > thank you! I am happy to raise my score at this point and I will update my review accordingly.
> > >
> > > Best,
> > > Reviewer 9nxn

---

### Official Review · Reviewer_fHDg · 2021-07-16

**Rating:** 6
**Confidence:** 3

**Summary:**

This paper proposes a MILP-based method for lossless compression of neural networks by removal of ReLU neurons whose output is characterized by a linear function, which can be merged or removed while inducing no change in functionality of the network.

**Limitations And Societal Impact:**

Limitations are largely discussed, as is societal impact.

**Main Review:**

Strengths:
  - Paper is largely well written and comprehensive.
  - Paper furthers a relatively unexplored area of neural network compression (fully lossless compression).
  - Work seems to be largely based on the cited work of Serra, et al, but shows improvement over that method in terms of runtime requirements and capability.

Weaknesses:
  - Method does not appear to be feasible when discussing very large networks.
  - Further analysis on input selection could improve the paper.

**Time Spent Reviewing:**

4

---

> ### Author Response · Authors · 2021-08-10
> **Initial response to R1 [fHDg]**
>
> We appreciate your praise and fair assessment of our work. It is indeed challenging to reconcile the application to large neural networks with advancing a relatively unexplored topic. While we do not believe that a single paper can handle it all, we have identified some promising possibilities for the two points that you have raised. In one way or another, these points were also identified by other reviewers, and thus included in our main response.
>
> For your convenience, we copy them below following the items that you have listed:
>
> **Method does not appear to be feasible when discussing very large networks**
>
> One possible simplification of our method would be to use only the preprocessing step described in Section 5.3 to determine which neurons to remove, since that step only takes a few seconds. Note that the preprocessing step is in principle intended to identify neurons that are not stable in order to avoid spending further time on them, but we can conversely assume that all the other neurons are stable at the cost of removing more than we should.
>
> By using preprocessing alone, our experiments would identify on average 31.93% potentially stable neurons in MNIST classifiers, 40.86% in CIFAR-10 classifiers, and 41.98% in CIFAR-100 classifiers. Among those neurons, only a few are actually not stable when evaluated with the test set. In relative terms, which we calculate as the number of not stable neurons with respect to the test set divided by the number of stable neurons with respect to the training set, we would have removed 1.16% more neurons that we should for MNIST,  0.60% for CIFAR-10, and 1.19% for CIFAR-100.
>
> **Further analysis on input selection could improve the paper**
>
> As a preliminary investigation, we have tested the effect of bounding the sum of all the MNIST inputs to be within the minimum and maximum observed values. In particular, we have constrained the sum of all inputs to be within the interval [15, 320] instead of [0, 784].  We believed that this approach would be preferable to constraining the value of individual inputs, since that would have affected the output upon rotation and translation.
>
> Note that this constraint is equivalent to imposing a prior on the number of foreground pixels on the digits to be within a range. Along the same lines, global priors that jointly act on all the pixels have been used in computer vision papers (e.g. label costs in Delong et al. IJCV-2012) in the pre-deep learning era.
>
> With this change we have obtained a better runtime in 69.6% of the cases after excluding those in which the time limit of 10,800 seconds was reached with and without this change. By restricting the analysis to the cases in which the time limit has not been exceeded either before or after the change, the geometric mean of the runtimes goes down by 17.7%.

---

### Author Response · Authors · 2021-08-10
**General response**

We appreciate the careful evaluation and useful suggestions provided by the reviewers. Thank you for capturing the intent of our work and also understanding the challenge that it entails to ensure that the compressed network is functionally equivalent to the original network while conservatively assuming that any input within the bounds is valid. We thank all the reviewers for considering our submission to be clear, well-written, and comprehensive. We also thank R2 for considering our submission to be insightful and impactful. We address some common points raised by the reviewers below, and then specific points directly to each reviewer. Please let us know if you have follow up questions.

### Application of method to larger networks (R1 [fHDg] and R4 [xWxW]):

One possible simplification of our method would be to use only the preprocessing step described in Section 5.3 to determine which neurons to remove, since that step only takes a few seconds. Note that the preprocessing step is in principle intended to identify neurons that are not stable in order to avoid spending further time on them, but we can conversely assume that all the other neurons are stable at the cost of removing more than we should.

By using preprocessing alone, our experiments would identify on average 31.93% potentially stable neurons in MNIST classifiers, 40.86% in CIFAR-10 classifiers, and 41.98% in CIFAR-100 classifiers. Among those neurons, only a few are actually not stable when evaluated with the test set. In relative terms, which we calculate as the number of not stable neurons with respect to the test set divided by the number of stable neurons with respect to the training set, we would have removed 1.16% more neurons that we should for MNIST,  0.60% for CIFAR-10, and 1.19% for CIFAR-100.

### Limiting exact compression to more likely inputs (R1 [fHDg] and R2 [9nxn]):

As a preliminary investigation, we have tested the effect of bounding the sum of all the MNIST inputs to be within the minimum and maximum observed values. In particular, we have constrained the sum of all inputs to be within the interval [15, 320] instead of [0, 784].  We believed that this approach would be preferable to constraining the value of individual inputs, since that would have affected the output upon rotation and translation.

Note that this constraint is equivalent to imposing a prior on the number of foreground pixels on the digits to be within a range. Along the same lines, global priors that jointly act on all the pixels have been used in computer vision papers (e.g. label costs in Delong et al. IJCV-2012) in the pre-deep learning era.

With this change we have obtained a better runtime in 69.6% of the cases after excluding those in which the time limit of 10,800 seconds was reached with and without this change. By restricting the analysis to the cases in which the time limit has not been exceeded either before or after the change, the geometric mean of the runtimes goes down by 17.7%.

### A closer look at Figure 3 (R2 [9nxn] and R4 [xWxW]):

When used in moderate amounts, regularization improves accuracy and very often that also coincides with exact compression being possible. We believe that the negative impression with the larger plots is due to how far we have stretched the experiments on 2x100 classifiers to show that more regularization leads to even more compression, which is actually just an extra but minor result of our experiments. That is particularly the case for the smallest architectures tested, and two of those (h and o) are among the exceptions to the rule that regularization improves accuracy and leads to exact compression.

We observe regularization improving accuracy in 17 of the 21 plots in Figure 3. In 14 of those cases, we can reduce the size of these more accurate networks. Those include all architectures for MNIST (a to g), the architecture with width 400 for CIFAR-100 (r), and all the architectures with 3 hidden layers for CIFAR-10 and CIFAR-100 (j, l, n, q, s, u).

In order to make that more evident, we will change the background color of every plot in the range of regularization values for which (a) the accuracy is better than with no regularization and (b) some amount of exact compression is observed. We can also make these regions more evident by not plotting the results for very large amounts of regularization, if the reviewers think that this would make the main message more clear to the readers.

### Extension to CNNs (R2 [9nxn] and R4 [xWxW]):

Following the suggestion of R2 (9nxn), we have tested our method with the LeNet architecture on CIFAR-10 using the preprocessing step as a predictor of neuron stability to make it more scalable. We note that in this case we would only use our method as a sparsification technique to mask stably inactive zeros due to parameter sharing.

When no regularization is used and the test accuracy on CIFAR-10 is around 68.7% before pruning,  we have found that an average of 10.98% of the stably inactive neurons can be masked as 0. With an L1 regularization of 0.000175, test accuracy on CIFAR-10 is around 70.02% before pruning while an average of 11.86% of the stably inactive neurons can be masked as 0. In comparison to the case of MLPs, we observed more variability on the number of stable neurons across networks trained with the same amount of regularization, which we believe is due to weight sharing. Similar to the case of MLPs, the proportion of neurons that are stable in the training set but not stable in the test set is relatively small: 1.06% on average. Moreover, we have observed that pruning those extra neurons has a zero net effect on accuracy for regularization values in the interval [0, 3.0e-04].

On a final note, we emphasize that masking 10% of the neurons is more strict than masking 10% of the parameters as done for lossy compression. Furthermore, masking 10% of the neurons does not prevent someone from sparsifying the CNN even further: our method is merely identifying a set of neurons - and corresponding parameters - that can be ignored for not being relevant. We believe that our method could be used in conjunction with conventional sparsification techniques in order to decompose the pruning operations of those into a lossless and a lossy component.

### Changes to paper and supplementary material (R2 [9nxn] and R3 [nANE]):

We appreciate that R2 and R3 analyze the supplemental material in detail, and we agree with the recommendations of two reviewers, which we have combined as follows:
- Bring Algorithm 2 from the supplemental material to the paper.
- Move Algorithm 1 and related theoretical results in Section 5.4 to supplemental material.
- Add comments to Algorithm 1 to make it easier to follow.
- Use space gained in the paper to add some of the explanations and experimental results produced for the rebuttal.

### Comments on side-effects of inexact pruning (R2 [9nxn] and R4 [xWxW]):

Regarding the clarification asked by R4 in page 1, the comment in line 34 refers to prior statement in line 27:

“They may also disproportionately affect some inputs more than others [42].”

We understand that this explanation is too brief, and we intend to rephrase it as follows:

“They may also lead to models in which the relative accuracy for some classes is more affected than that of other classes [42].”

In line 34, we will replace “some inputs” with “some classes”.

Regarding the correction asked by R2 in page, we propose rephrasing it as follows:

“Large models are resource-intensive for both training as well as inference. In contrast to approximate methods, our exact model compression algorithms can help deep learning practitioners to save computational time and resources without worrying about any loss in performance. That helps preventing the documented side effect of disproportionally degrading performance for some classes more than for other classes when the indicator of a successful compression is the overall performance, which could also lead to fairness issues as discussed in the relevant literature [SOURCES].”

In addition to citing the existing reference [42], we will add the following papers:

https://arxiv.org/abs/2009.09936

https://arxiv.org/abs/2010.03058c

---

### Author Response · Authors · 2021-08-11
**Additional comments on the case of CNNs**

We just would like to note that we have extended our answer regarding the application to CNNs by editing the original answers. The reviewers may have not received an email notification about that, but we kindly ask that you consider the current version of the answers.

---

### Decision · Program_Chairs · 2021-09-27

**Decision:**

Accept (Poster)

**Comment:**

All the reviewers have agreed that the submission is original and interesting with clear/practical contributions to the domain. Authors’ detailed and informative responses also addressed most of the major concerns and some of the reviewers accordingly increased their scores. Given the overwhelming positive opinions, I am recommending an acceptance.